# Gasdermin D promotes influenza virus-induced mortality through neutrophil amplification of inflammation

Samuel Speaks [1,4], Matthew I. McFadden[1,2,4], Ashley Zani[1,2,4], Abigail Solstad[1], Steve Leumi[1,2], Jack E. Roettger [1,2], Adam D. Kenney[1,2], Hannah Bone [1,2], Lizhi Zhang[1,2], Parker J. Denz [1,2], Adrian C. Eddy[1,2], Amal O. Amer [1,2], Richard T. Robinson [1,2], Chuanxi Cai[3], Jianjie Ma[3], Emily A. Hemann[1,2], Adriana Forero [1,2] ✉ & Jacob S. Yount [1,2] ✉

Influenza virus activates cellular inflammasome pathways, which can be both beneficial and detrimental to infection outcomes. Here, we investigate the function of the inflammasome-activated, pore-forming protein gasdermin D (GSDMD) during infection. Ablation of GSDMD in knockout (KO) mice (*Gsdmd*−/−) significantly attenuates influenza virus-induced weight loss, lung dysfunction, lung histopathology, and mortality compared with wild type (WT) mice, despite similar viral loads. Infected *Gsdmd*−/− mice exhibit decreased inflammatory gene signatures shown by lung transcriptomics. Among these, diminished neutrophil gene activation signatures are corroborated by decreased detection of neutrophil elastase and myeloperoxidase in KO mouse lungs. Indeed, directly infected neutrophils are observed in vivo and infection of neutrophils in vitro induces release of DNA and tissue-damaging enzymes that is largely dependent on GSDMD. Neutrophil depletion in infected WT mice recapitulates the reductions in mortality, lung inflammation, and lung dysfunction observed in *Gsdmd*−/− animals, while depletion does not have additive protective effects in *Gsdmd*−/− mice. These findings implicate a function for GSDMD in promoting lung neutrophil responses that amplify influenza virus-induced inflammation and pathogenesis. Targeting the GSDMD/neutrophil axis may provide a therapeutic avenue for treating severe influenza.

Influenza A virus (IAV) infection remains a threat to global public health with seasonal epidemics affecting nearly 10% of the world's population annually and emergent global pandemics remaining an ever-present threat[1,2]. Exuberant inflammatory immune responses and tissue damage often characterize severe IAV infections. Indeed, IAV is known to trigger multiple inflammatory pathways, including the activation of cellular inflammasomes through several mechanisms[3]. However, whether inflammasome activation is beneficial or detrimental to the outcome of viral infections is context dependent[4–9], which could indicate that distinct pathways and molecules downstream of inflammasome activation have divergent effects.

The NLRP3 inflammasome is amongst the best characterized inflammasomes in the context of IAV infection and its activation is a double-edged sword, driving both beneficial effects and pathological inflammation[3,6]. NLRP3 knockout (KO) mice exhibit accelerated death upon IAV infection, a phenotype that is associated with a decrease in

[1]Department of Microbial Infection and Immunity, The Ohio State University, Columbus, OH, USA. [2]Infectious Diseases Institute, The Ohio State University, Columbus, OH, USA. [3]Department of Surgery, Division of Surgical Science, University of Virginia, Charlottesville, VA, USA. [4]These authors contributed equally: Samuel Speaks, Matthew I. McFadden, Ashley Zani. ✉e-mail: Adriana.Forero@osumc.edu; Jacob.Yount@osumc.edu

protective proinflammatory cytokine secretion[5]. On the other hand, blockade of the NLRP3 inflammasome via chemical inhibition at day 7 post infection is beneficial during influenza virus infection and was also correlated with decreased production of inflammatory cytokines and chemokines[6,10–12]. In addition, we have previously shown dramatically reduced morbidity, mortality, and lung inflammation in IAV-infected mice treated with an experimental therapeutic protein, recombinant human Mitsugumin 53 (MG53, also known as TRIM72[13]), which was associated with decreased NLRP3 inflammasome levels and activity[14]. Together, these prior studies suggest that, while some inflammasome functions may be beneficial, excessive inflammasome activation contributes to pathology and death during IAV infection.

Gasdermin D (GSDMD) is an effector protein cleaved by inflammasome-activated caspases, allowing the liberated N-terminal GSDMD fragment to assemble into oligomers that form pores in the plasma membrane[15,16]. GSDMD pores mediate release of specific pro-inflammatory cytokines, namely IL-1 and IL-18[17–19], which can promote leukocyte recruitment and viral clearance[20–22], but unresolved pore formation can lead to pyroptotic cell death[15,23–25]. Despite these important functions, the contributions of GSDMD to viral pathogenesis have been only minimally investigated. In vitro, GSDMD was shown to be non-essential for macrophage death induced by infection with a mouse-adapted influenza virus strain, while effects on cytokine and chemokine responses were not examined[8]. GSDMD has been rarely investigated in viral pathogenesis studies in animal models[4,9,26]. Given its role as a driver of inflammation and pyroptosis, the role of GSDMD in driving inflammation during viral infection in vivo warrants further investigations.

In the present study, we explore the function of GSDMD in IAV infections using *Gsdmd⁻/⁻* mice, which experience decreases in lung inflammation, lung dysfunction, and overall mortality during this viral infection, as well as dampened neutrophil activation and lung inflammation compared to wild type (WT) mice. The decreases in inflammation in the absence of GSDMD do not affect virus loads or hinder recovery from infection. Our study identifies GSDMD as a potential target for decreasing pathological neutrophil activities and the severity of influenza virus infections.

## Results

### Reduced morbidity and mortality in IAV infected *Gsdmd⁻/⁻* mice

To investigate the roles of GSDMD during IAV infection, we infected female WT and *Gsdmd⁻/⁻* mice with IAV strain A/Puerto Rico/8/34 (H1N1) (referred to hereafter as PR8). Female mice were primarily used in our studies because of their increased severity of PR8 infection compared to male mice[27]. We first examined lungs for GSDMD levels, including cleaved GSDMD N-terminal fragment indicative of

activation, at day 7 post infection via Western blotting. Cleaved GSDMD was detected in the lungs of infected WT mice, as expected[14], whereas only full-length GSDMD could be seen in lungs from mock-infected WT mice (Fig. 1A). GSDMD could not be detected in lungs from infected *Gsdmd⁻/⁻* mice, confirming genetic ablation in these animals (Fig. 1A). *Gsdmd⁻/⁻* mice lost significantly less weight than their WT counterparts (Fig. 1B), though it is important to note that the sickest WT mice succumbed to infection, masking the full extent of the protective benefits of GSDMD deficiency at later timepoints. Indeed, 60% of WT mice succumbed to infection or met humane endpoint criteria of greater than 30% weight loss, compared to roughly 10% of *Gsdmd⁻/⁻* animals, demonstrating a protective effect when GSDMD is absent (Fig. 1C). We next evaluated whether GSDMD impacts viral replication. We found that virus titers in the lungs of WT and KO mice were similar (Fig. 1D), indicating that GSDMD contributes to the morbidity and mortality of IAV infections without directly affecting virus levels.

### GSDMD increases lung pathology and dysfunction during IAV infection

To further explore the protective effects provided by GSDMD deficiency during IAV infection, we measured enhanced pause (PenH) values, which provides a surrogate indicator of airway resistance, in WT and KO mice throughout a timecourse of infection by whole body plethysmography[28]. PenH increased in both WT and KO mice following infection and peaked in both groups at day 7 post infection (Fig. 2A). However, the magnitude of PenH values was significantly lower in the *Gsdmd⁻/⁻* mice (Fig. 2A), revealing that the absence of GSDMD during IAV infection is beneficial to lung function. We further examined lung sections via hematoxylin and eosin staining at day 7 post infection. Although all infected lung sections showed areas of cell infiltration and consolidation, lungs from WT mice exhibited more severe pathology, with thickened alveolar septa, cellular accumulation, and less open airspace as compared to *Gsdmd⁻/⁻* lungs (Fig. 2B). Areas of consolidation versus open airspace were quantified as a measure of pathology in individual mice[9,29]. *Gsdmd⁻/⁻* mice indeed showed significantly decreased lung pathology via this unbiased quantification method as compared to WT mice (Fig. 2C). We next examined whether the protective benefits of the loss of GSDMD extended to male mice. We observed that male mice had a less severe infection with moderate weight loss and no fatalities, and that *Gsdmd⁻/⁻ mice*, on average, experienced less weight loss during the infection than WT mice (Supplementary Fig 1A). Furthermore, similarly to female counterparts (Fig. 2A), male *Gsdmd⁻/⁻* mice exhibited a statistically significant benefit in terms of lung function as indicated by PenH values (Supplementary Fig 1B). Our results overall demonstrate that the absence of

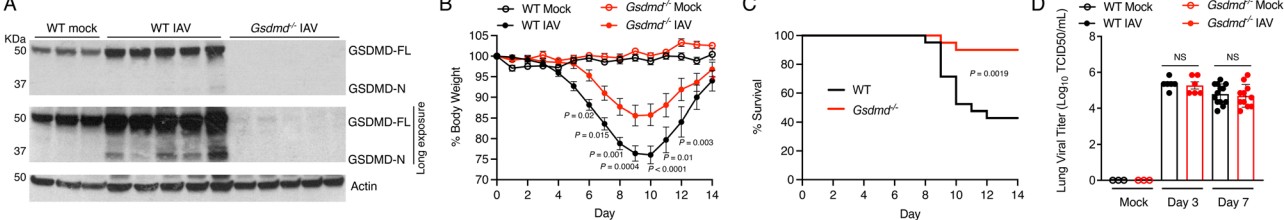

**Fig. 1 | GSDMD promotes inflammation and morbidity following IAV infection. A-D** WT and *Gsdmd⁻/⁻* mice were intranasally infected with 50 TCID50 of IAV strain PR8. **A** Western blot of lung lysates taken at day 7 post infection. Each lane represents lysate from an individual mouse. **B** Weight loss measurements (each dot is an average of individual mouse weights normalized to 100% relative to day 0, error bars indicate SEM, Mocks: n = 4 WT and *Gsdmd⁻/⁻* from a single experiment, IAV-infected: n = 21 WT, n = 20 KO from 3 independent experiments, p values determined by two-way ANOVA and Bonferroni multiple comparisons test, only statistical comparisons between infected groups are shown). **C** Survival curve from 3

independent experiments as in **B** (p value by Log-rank Mantel-Cox test). **D** Viral titers from lung homogenates from mock-infected, day 3 post infection (from 1 experiment) or day 7 post infection (from 2 independent experiments) (each dot represents an individual mouse, n = 3 mocks, n = 6 day 3 WT and KO, n = 11 for day 7 WT and *Gsdmd⁻/⁻*, error bars indicate SEM). NS, not significant by two-way ANOVA with Tukey's Multiple Comparisons Test, only statistical comparisons for WT versus *Gsdmd⁻/⁻* on days 3 and 7 are shown. Source data are provided in the Source Data file.

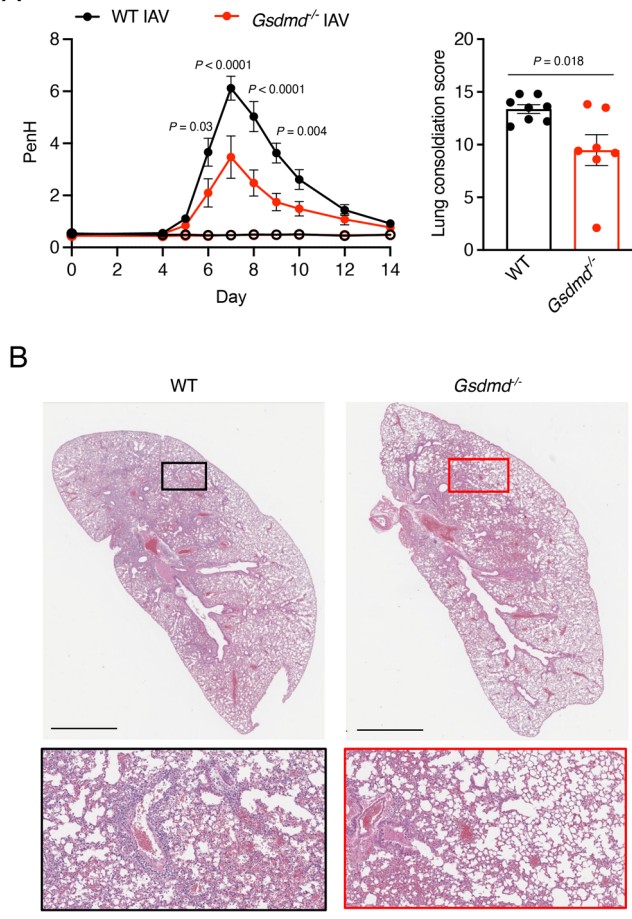

**Fig. 2 | GSDMD exacerbates lung dysfunction and pathology following IAV infection. A–C** WT and *Gsdmd⁻/⁻* mice were intranasally infected with 50 TCID50 of IAV strain PR8. **A** Daily whole body plethysmography enhanced pause (PenH) measurements (each dot represents averages from multiple mice, Mocks: *n* = 4 WT and *Gsdmd⁻/⁻* (from one experiment), IAV-infected: *n* = 18 WT, *n* = 17 *Gsdmd⁻/⁻* (from 3 independent experiments), error bars indicate SEM, p values determined by two-way ANOVA with Bonferroni Multiple Comparisons Test comparing WT IAV versus *Gsdmd⁻/⁻* IAV, other comparisons not shown). **B** Representative H&E images for WT and *Gsdmd⁻/⁻* mice at day 7 post-infection. Insets show magnified areas of cell infiltration and inflammation. Scale bars represent 2 mm. **C** Quantification of consolidated tissue versus open airspace in H&E staining for entire lung sections of multiple IAV-infected animals (each dot represents an individual mouse, *n* = 8 WT and *n* = 7 *Gsdmd⁻/⁻* mice, error bars indicate SEM (*p < 0.05 by two-tailed unpaired t-test). Source data are provided in the Source Data file.

GSDMD results in an attenuation of IAV-induced lung pathology in both female and male mice. Subsequent experiments were performed using the more severe female mouse infection system in order to have a robust dynamic range for observing phenotypic and mechanistic differences between WT and *Gsdmd⁻/⁻* mice.

### *Gsdmd⁻/⁻* mice have decreased inflammatory transcriptional responses to IAV infection

To better understand the underlying mechanisms contributing to differences in survival and lung damage between WT and *Gsdmd⁻/⁻* animals, we compared global gene transcriptional responses of WT and *Gsdmd⁻/⁻* mice following IAV infection. Quantitative measurement of total lung mRNA expression at day 7 post infection showed distinct transcriptional profiles induced in WT and *Gsdmd⁻/⁻* animals as revealed by principal components analysis (Fig. 3A). Differential gene expression analysis (fold change |3 |, *p*-adj <0.01) captured 812 genes that were increased in infected WT mice compared to infected

*Gsdmd⁻/⁻* mice, while expression of 447 genes was increased in *Gsdmd⁻/⁻* relative to WT mice, for a total of 1259 differentially expressed genes (Fig. 3B, Supplementary Data 1). Hierarchical clustering of gene expression revealed gene clusters that were highly upregulated (yellow cluster) and moderately upregulated (red cluster) in infected WT mice relative to *Gsdmd⁻/⁻* mice (Fig. 3C). Gene ontology enrichment revealed that both highly and moderately upregulated genes corresponded to biological processes involved in inflammatory and antiviral responses (Fig. 3D). Genes found to be repressed in WT infected mice relative to *Gsdmd⁻/⁻* mice (orange cluster), such as *Kcnh2, Scn5a*, and *Sgcg*, were found to associate with muscle function. Gene set enrichment analysis followed by network analysis further confirmed that biological processes associated with inflammation and cytokine signaling were dampened by the absence of GSDMD in IAV infection (Fig. 3E).

### Loss of GSDMD reduces inflammation during IAV infection

Further analysis of RNA sequencing data illustrated that gene expression linked to inflammatory defense responses to virus infection was decreased in *Gsdmd⁻/⁻* lungs (Fig. 4A). This gene set included interferons (IFNs), IFN-stimulated chemokines, and classical antiviral IFN-stimulated genes such as MX and OAS family genes (Fig. 4A, Supplementary Data 2). We utilized published immune-cell-specific databases (PanglaoDB) to conduct functional enrichment analysis of our differentially expressed genes and discovered significant decreases in genes associated with myeloid cells in the lungs of *Gsdmd⁻/⁻* mice (Supplementary Fig 2A). This was accompanied by observed enrichment of biological pathways involved in neutrophil chemotaxis in WT versus *Gsdmd⁻/⁻* samples as identified by GO Biological Process analysis (Fig. 4B, Supplementary Data 2), as well as by REACTOME and IPA Canonical Pathway analyses (Supplementary Fig 2B, C). These neutrophil-associated genes included a number of cytokines, chemokines, and receptors, with *Ccl1* being the most significantly downregulated gene identified (Figs. 3B, 4B). These results may suggest a decrease in neutrophil numbers or decreased activation of recruited neutrophils[30]. To confirm these transcriptomic results, we performed qRT-PCR on lung RNA from additional cohorts of mock- or IAV-infected animals to measure expression of a panel of inflammatory genes. We broadly confirmed a decrease in inflammatory genes, including *Ccl1, Cxcl9, Tnf*, and *Ifnb1*, among others, in infected *Gsdmd⁻/⁻* versus WT mice, while no baseline differences were observed in the absence of infection (Fig. 4C). We further confirmed this attenuation of inflammatory cytokine and chemokine gene expression by measurement of secreted factors. We observed that the chemokine CCL1 was significantly decreased in *Gsdmd⁻/⁻* versus WT lungs (Fig. 4D), consistent with the highly significant changes observed for this chemokine in our gene expression assays (Fig. 4B, C). *Gsdmd⁻/⁻* mice also had reduced levels of pro-inflammatory cytokines, including IL-6, IFNβ, TNF, IL-1β, and IL-18 (Fig. 4D). These results were consistent with the established role of GSDMD in facilitating release of IL-18 and IL-1β[17–19] and with our RNA sequencing results that indicated the down-modulation of IL-1 signaling in KO lungs (Fig. 3D). Overall, these RNA sequencing, qRT-PCR, and ELISA results broadly identified decreased inflammatory responses during IAV infection in *Gsdmd⁻/⁻* mice.

Because GSDMD is known to mediate cytokine release and pyroptosis in macrophages, we investigated the response of human THP-1 macrophages to PR8 or an H3N2 IAV seasonal isolate with or without shRNA targeting of GSDMD levels. We observed a decrease in IL-1β, IL-6, TNF, and IFNβ levels in the supernatants of GSDMD knockdown (KD) macrophages despite comparable levels of infection as indicated by viral nucleoprotein levels in Western blots (Supplementary Fig. 3A–C). We also observed increased levels of cleaved PARP1 in WT and GSDMD KD cells after infection indicating that cell death induced by infection was not broadly prevented by GSDMD depletion (Supplementary Fig 3A), though lactate dehydrogenase release was decreased in

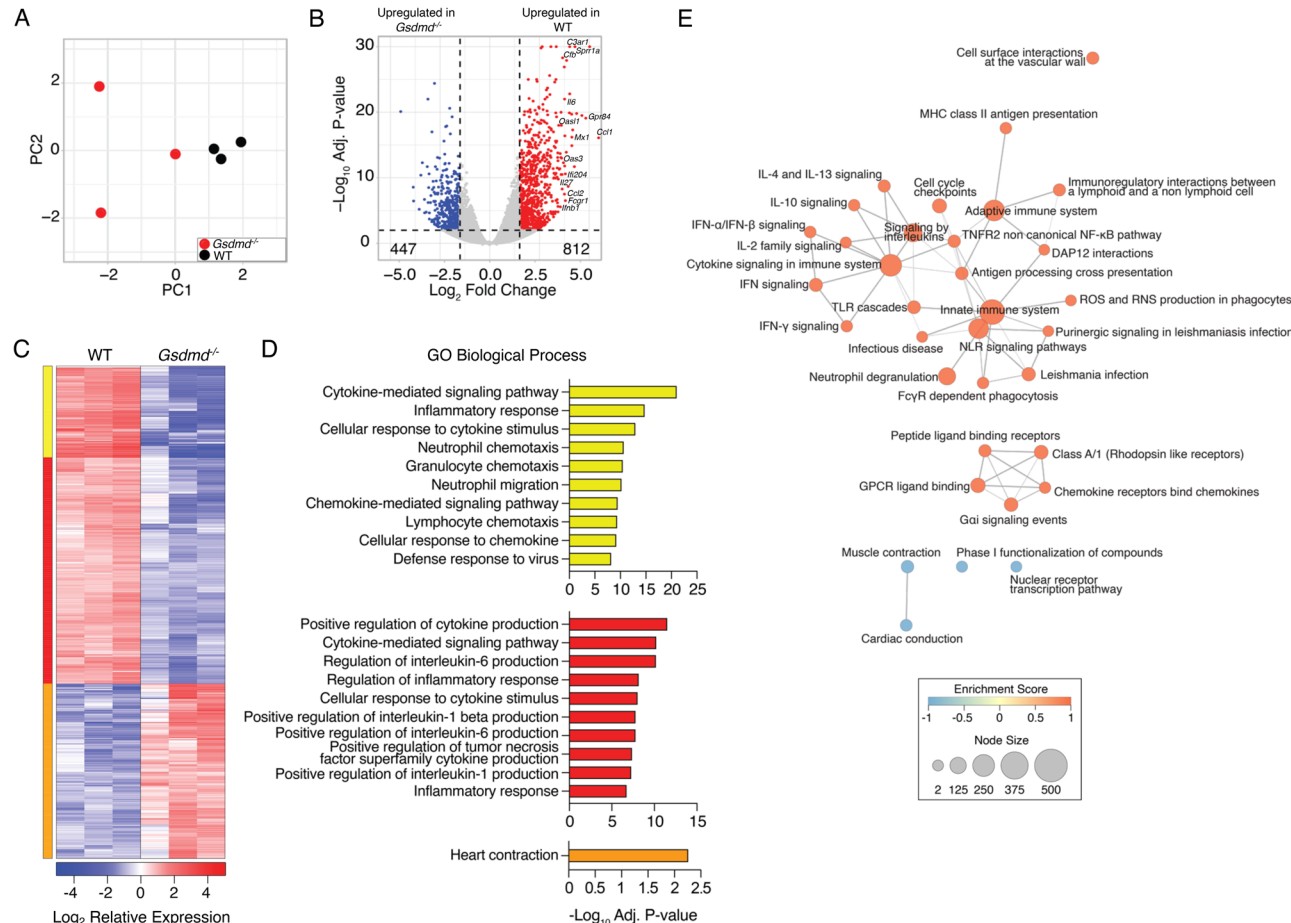

**Fig. 3 | GSDMD promotes inflammatory gene expression programs in IAV infection. A–E** WT and *Gsdmd*[-/-] mice were infected with IAV strain PR8 at a dose of 50 TCID50. RNA was extracted from *n* = 3 WT and *Gsdmd*[-/-] mouse lungs at day 7 post infection and subjected to RNA sequencing. **A** Principal component analysis comparing WT and *Gsdmd*[-/-] lung RNA sequencing results. Each dot represents an individual mouse. **B** Volcano plot of differential gene expression (lfc |1.5|, adj *p*-value < 0.01 determined using DEseq2) comparing WT vs *Gsdmd*[-/-]. Red, 812 genes upregulated in WT vs KO; Blue, 447 genes downregulated in WT vs *Gsdmd*[-/-]. **C** Hierarchical clustering of differentially expressed genes. Heatmap represents relative gene expression where red indicates genes with upregulated expression and blue indicates genes with downregulated expression, with each column representing an individual mouse. Cluster color indicates genes with similar patterns of expression in infected WT mice compared to infected *Gsdmd*[-/-] mice. **D** Gene ontology analysis for genes within each cluster. Bar graphs represent the top 10 enriched GO terms enriched within each cluster from **C** as indicated by color. Bar length represent the -log10 adjusted *p*-value for significantly enriched pathways (-Log10 adj *p*-value > 1.3 by topGO enrichment analysis for gene ontology). **E** Network of GO Biological Process terms enriched by gene set enrichment analysis. Node size represents number of genes within each pathway. Edges represent the number of shared genes across GO Biological Process terms. Color indicates the Enrichment Score where orange indicates a positive score and blue indicates a negative score for transcriptional signatures derived from WT infected lungs relative to *Gsdmd*[-/-] lungs.

GSDMD KD cells (Supplementary Fig 3D). Our data overall show that IAV-induced inflammatory responses are dampened in the absence of GSDMD.

**Loss of GSDMD reduces neutrophil activation programs during IAV infection**

Additional interrogation of our global gene expression data identified significant decreases in gene sets associated with neutrophil extracellular trap formation, NOD-like receptor signaling, and neutrophil degranulation in IAV infected *Gsdmd*[-/-] versus WT mice (Fig. 5A, B), suggesting that reduced neutrophil activities may play a role in the attenuated pathogenesis observed in *Gsdmd*[-/-] mice. Thus, we examined whether GSDMD was involved in neutrophil recruitment to the lungs during infection using flow cytometry (Supplementary Fig 4). Surprisingly, we found that both WT and *Gsdmd*[-/-] mice exhibited robust neutrophil recruitment to the lungs by day 7 post infection with no statistical difference observed between the genotypes in terms of numbers of neutrophils per lung (Fig. 5C) or percent of neutrophils relative to total CD45[+] immune cells (Supplementary Fig 5). We also

observed similar recruitment of other innate immune cell populations, such as eosinophils, macrophages, or natural killer cells into the lungs, and likewise, recruitment of adaptive CD4 and CD8 T cells and B cells was not impaired in *Gsdmd*[-/-] animals (Supplementary Fig 5, Supplementary Fig 6A). We additionally measured immune cell recruitment to the lungs at day 3 post infection and detected minimal cell infiltration and no differences between WT and *Gsdmd*[-/-] mice (Fig. 5C, Supplementary Fig 5, Supplementary Fig 6A). Staining for the hematopoietic immune cell marker CD45 in lung sections at day 7 post infection confirmed recruitment of lymphocytes to the lungs in both WT and KO animals (Supplementary Fig 6B). We thus hypothesized that despite similar numbers of neutrophils in the lungs of *Gsdmd*[-/-] mice, neutrophil functionality may be decreased. To test this, we measured levels of neutrophil elastase, a proteolytic enzyme released by activated neutrophils that degrades extracellular matrix and promotes neutrophil extracellular trap formation[31–33]. This indicator of activated neutrophils was significantly decreased in the lungs of infected *Gsdmd*[-/-] mice as compared to WT tissue (Fig. 5D). We similarly measured myeloperoxidase, a molecule released by azurophilic

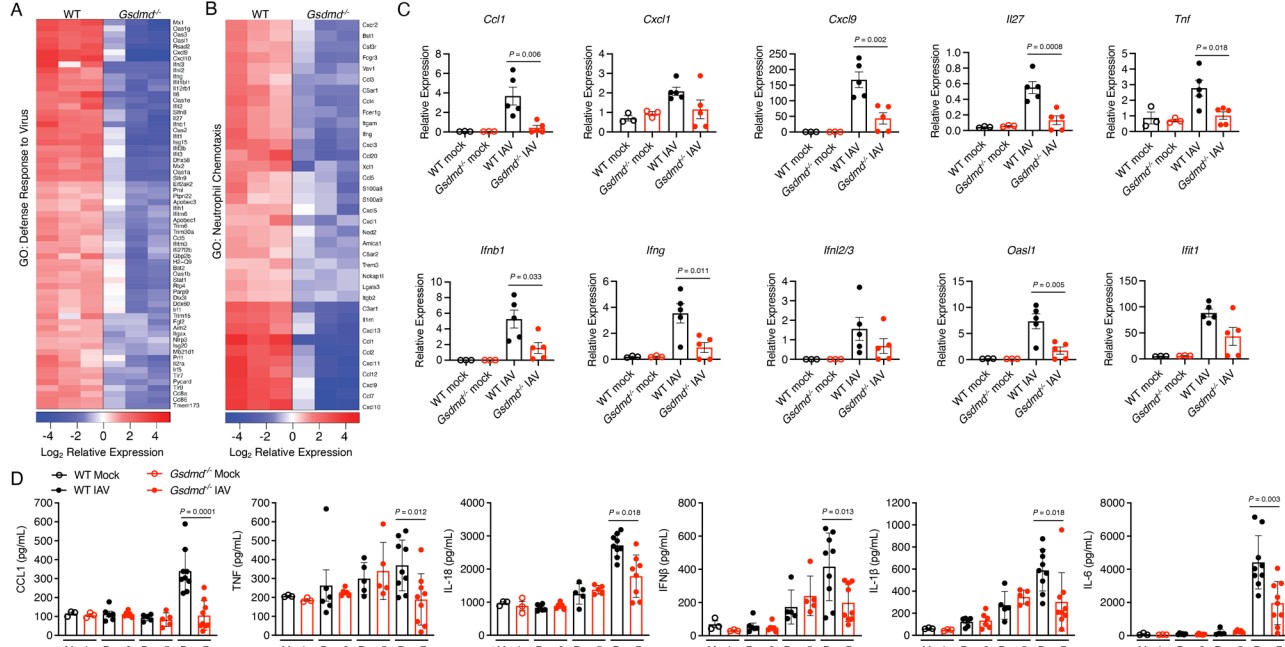

**Fig. 4 | IAV-induced inflammation is diminished in *Gsdmd*⁻/⁻ mice. A, B** RNAseq results as in Fig. 3 were further analyzed. **A** Heat map for top differentially regulated genes involved with Response to Virus as defined by GO Biological Process. **B** Heatmap for top differentially regulated genes involved with neutrophil chemotaxis as defined by GO Biological Process. **C** Relative transcript quantification from targeted genes in the lungs of WT and *Gsdmd*⁻/⁻ mice at day 7 post infection with 50 TCID50 IAV strain PR8 via qRT-PCR (each dot represents a single mouse, error bars indicate SEM, Mocks: n = 3, IAV-infected: *n* = 5 from 1 experiment, samples are independent from those used for RNAseq, *p* values determined by one-way ANOVA followed by Tukey's multiple comparisons test, only comparisons between infected WT and *Gsdmd*⁻/⁻ are shown). **D** ELISA quantification of TNF, IL-6, IFNβ, IL-1β, IL-18 or CCL1 levels in lung homogenates of mock-infected, day 3 post infection (from 1 experiment), day 5 post infection (from 1 experiment), or day 7 post infection (from 2 independent experiments) (each dot represents a single mouse, Mocks: n = 3, IAV day 3: *n* = 6, IAV day 5: *n* = 5, IAV day 7: *n* = 9, error bars indicate SEM, *p* values determined by two-way ANOVA followed by Tukey's multiple comparisons test, only comparisons between day 7 infected WT and *Gsdmd*⁻/⁻ are shown). Source data are provided in the Source Data file.

neutrophil granules that catalyzes production of reactive oxygen intermediates known to contribute to tissue damage, and found that it was also decreased in *Gsdmd*⁻/⁻ lung tissue (Fig. 5D). Thus, products of activated neutrophils present in lungs during IAV infection were decreased in the absence of GSDMD.

To further examine roles for GSDMD in neutrophils, we next infected primary neutrophils enriched from the bone marrow of WT or *Gsdmd*⁻/⁻ mice (enrichment shown in Supplementary Fig 7). Ethidium homodimer-1 fluorescence in WT neutrophil media increased over time following infection, indicating that DNA release characteristic of NETosis occurred in response to virus exposure (Fig. 5E, F). Interestingly, minimal DNA staining was observed over the same time course for *Gsdmd*⁻/⁻ neutrophils (Fig. 5E, F). Corroborating these results, we observed that release of neutrophil elastase and myeloperoxidase into the cell supernatants following infection was largely dependent on GSDMD (Fig. 5G). Given that neutrophils were previously reported to be directly infected by influenza virus[34–36] and given that our in vitro results suggest a direct response of these cells to virus (Fig. 5E–G), we sought to confirm that neutrophils are infected by IAV in vivo. Using Cre recombinase-expressing PR8 in a Cre-inducible fluorescent reporter mouse[37–39], we observed infection of alveolar macrophages, a well characterized immune cell target of IAV, and also observed a population of neutrophils expressing the Cre-induced fluorescent reporter of infection (Supplementary Fig 8A). We further noted that the infected neutrophils showed higher MHCII surface levels than non-infected cells, highlighting neutrophil activation upon infection (Supplementary Fig 8B). These findings prompted us to re-evaluate neutrophil surface markers in our previously performed flow cytometry experiments from WT or *Gsdmd*⁻/⁻ lungs. We found that levels of Ly6G, CD11b, and Ly6C were unchanged in WT and *Gsdmd*⁻/⁻ neutrophils with or without infection (Supplementary Fig 9A, B). In contrast, average

MHCII mean fluorescence intensity was increased on lung neutrophils following IAV infection in both WT and *Gsdmd*⁻/⁻ mice (Supplementary Fig 9A, B). These results together uncover that neutrophils are activated upon IAV infection through GSDMD-dependent and -independent pathways. Importantly, our RNA sequencing results, measurements of neutrophil products in vivo, and examination of neutrophil activity in response to IAV in vitro each indicate a dependence on GSDMD for neutrophil functionality, including release of DNA and tissue-damaging enzymes, during IAV infection.

## Neutrophils increase inflammation and mortality in IAV infection

To probe whether dampened neutrophil activities could explain the decrease in influenza virus infection severity that we observed in the *Gsdmd*⁻/⁻ animals, we depleted neutrophils early in infection in WT mice and measured infection outcomes. Briefly, we treated mice with antibodies targeting neutrophils (α-Ly6G) or an isotype control antibody, starting at day 3 post infection through day 8 post infection (Fig. 6A). This treatment timeline was chosen to allow acute neutrophil responses to proceed uninhibited prior to blockade of an excessive, pathological response, and also because it represented a test of neutrophil depletion as a therapeutic strategy. We confirmed neutrophil depletion by measuring decreases in total neutrophil numbers, as well as relative percentage of neutrophils compared to the entire CD45⁺ immune cell population, in the lungs of α-Ly6G treated mice at day 5 post infection (Fig. 6B, Supplementary Fig 10A–C). The specificity of neutrophil depletion was further confirmed by analyzing total counts and percentages of eosinophils and alveolar macrophages, which were unchanged between the two treatment groups. (Supplementary Fig 10A–C). When examining morbidity between the cohorts of mice, the neutrophil depleted mice experienced significantly less influenza

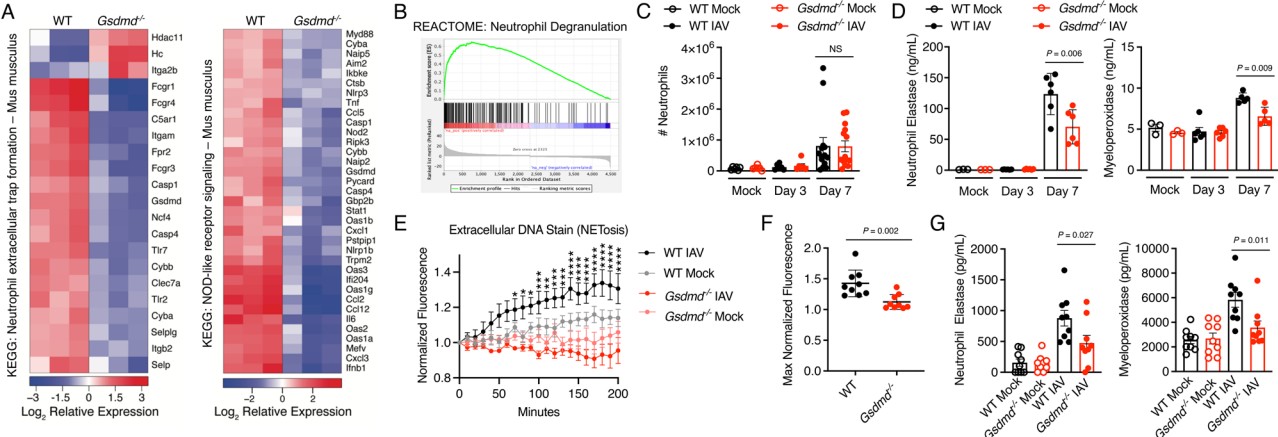

**Fig. 5 | IAV-induced neutrophil functions are decreased in the absence of GSDMD. A** RNAseq results as in Fig. 3 were analyzed to create heat maps for differentially regulated genes involved with neutrophil extracellular trap formation and NOD-like receptor signaling defined by KEGG pathway analysis. **B** REACTOME analysis of differentially expressed genes in WT versus *Gsdmd*$^{-/-}$ mice identified differences in genes associated with neutrophil degranulation. **C** Neutrophil infiltration into the lungs quantified by flow cytometry of mock-infected (from two independent experiments), day 3 post infection (from 1 experiment), or day 7 post infection (from 3 independent experiments) (each dot represents a single mouse, *n* = 6 mocks, *n* = 6 day 3, *n* = 14 day 7, error bars indicate SEM, NS, not significant by two-way ANOVA followed by Tukey's multiple comparisons test). **D** ELISA quantification of neutrophil elastase and myeloperoxidase levels in lung homogenates of mock-infected (from 1 experiment), day 3 post infection (from 1 experiment), or day 7 post infection (from 2 independent experiments) (each dot represents a

single mouse, *n* = 3 mocks, *n* = 6 day 3, *n* = 6 day 7, error bars indicate SEM, p values determined by two way ANOVA followed by Tukey's multiple comparisons test). **E** Relative extracellular DNA from bone marrow neutrophils was quantified via fluorescence intensity readings following IAV infection (PR8, MOI = 10, each data point is an average of *n* = 9 with cells derived from 3 mice for each genotype assayed in triplicate, error bars indicate SEM, **p* < 0.05, ***p* < 0.01, ****p* < 0.001, *****p* < 0.0001 determined by two-ANOVA followed by Bonferroni multiple comparisons test comparing WT IAV to *Gsdmd*$^{-/-}$ IAV, other comparisons not shown). **F** Maximum fluorescence intensity for each infected well as in **E** (*p*-value determined by two-tailed unpaired t-test). **G** Neutrophil products in supernatants from cells as in **E** were quantified via ELISA (*n* = 9 with cells derived from 3 mice assayed in triplicate, error bars indicate SEM, **p* values determined by one-way ANOVA followed by Tukey's multiple comparisons test). Source data are provided in the Source Data file.

virus-induced weight loss than the isotype control treated group (Fig. 6C). Remarkably, neutrophil depletion was completely protective against lethal infection, whereas we observed 60% lethality in isotype control-treated mice (Fig. 6D), a result that mirrored the protective effects seen in *Gsdmd*$^{-/-}$ mice (Fig. 1C).

Also similar to results comparing *Gsdmd*$^{-/-}$ and WT mice, viral titers in the lungs of neutrophil-depleted and isotype control-treated mice were not statistically different from one another at either day 5 or day 7 post infection (Fig. 6E). We further observed that levels of inflammatory cytokines, including IL-1β, IL-6, IFNβ, and TNF-α, were significantly decreased in the lungs of neutrophil-depleted mice compared to controls (Fig. 6F). Likewise, CCL1 levels were commensurately decreased following α-Ly6G treatment (Fig. 6F). These results suggest that neutrophils overall potentiate inflammation in IAV infected lungs. Lastly, we quantified neutrophil elastase in the lungs of α-Ly6G treated and isotype control treated mice to further confirm that neutrophil elastase serves as a marker of neutrophil presence and activity. Indeed, we saw decreased neutrophil elastase in the lungs of α-Ly6G treated mice (Fig. 6F), reinforcing our data identifying a similar change in *Gsdmd*$^{-/-}$ versus WT mice (Fig. 5D). These results demonstrate that neutrophils provide an overall potentiation of inflammation in IAV infected lungs.

To gain further insight into whether the protection provided by the loss of GSDMD during IAV infection was mediated by neutrophils, we depleted neutrophils from both WT and *Gsdmd*$^{-/-}$ animals during IAV infection (Supplementary Fig 11). We again observed that neutrophil depletion or loss of GSDMD protected mice from death and severe weight loss (Fig. 6G, H). Neutrophil-depleted *Gsdmd*$^{-/-}$ mice similarly survived the infection but did not show an added benefit over *Gsdmd*$^{-/-}$ alone in terms of weight loss (Fig. 6G, H). We also measured lung function via full body plethysmography of these mice. PenH indicative of airway resistance was increased by infection in WT isotype control-treated mice (Fig. 6I). WT mice with neutrophil depletion, *Gsdmd*$^{-/-}$ control mice, and *Gsdmd*$^{-/-}$ mice with neutrophil depletion

each showed less decline in lung function allowing a more rapid recovery to baseline lung function compared to WT control-treated mice (Fig. 6I). Importantly, similar to weight loss measurements, we did not observe an added benefit of combining *Gsdmd*$^{-/-}$ with neutrophil depletion (Fig. 6I). These plethysmography measurements demonstrate that neutrophil depletion protects lung function during IAV infection and are consistent with the notion that inflammatory, lung damaging effects of GSDMD during IAV infection are mediated to a significant extent by neutrophils. In sum, our results demonstrate that neutrophil activity in the lungs during influenza virus infection is largely dependent on GSDMD and that neutrophils amplify the inflammatory response and lung dysfunction during IAV infection.

## Discussion

We discovered that the inflammasome effector, GSDMD, has a detrimental impact on outcomes of influenza virus infection. In the absence of GSDMD, mice experienced reduced morbidity and mortality despite viral burden in the lungs being unaffected. Indeed, GSDMD contributed to lung inflammation, neutrophil activity, and lung dysfunction. The improved outcomes of *Gsdmd*$^{-/-}$ mice following IAV infection compared to WT mice corresponded to global transcriptional decreases in a multitude of tissue-damaging inflammatory pathways. Additionally, impaired neutrophil activity gene expression programs in the absence of GSDMD correlated with improved disease outcome, which we subsequently substantiated via exogenous depletion of these immune cells. While neutrophil depletion and genetic ablation of GSDMD are not equivalent, we note that they phenocopy each other in IAV infection outcomes in terms of decreased weight loss, improved survival, improved lung function, and decreased lung inflammation, each without affecting viral titers. These data are consistent with a model in which GSDMD-dependent neutrophil activities drive influenza virus pathogenesis. Additional data in support of this model include: 1) decreased levels of tissue damaging enzymes released by neutrophil degranulation (neutrophil elastase and myeloperoxidase)

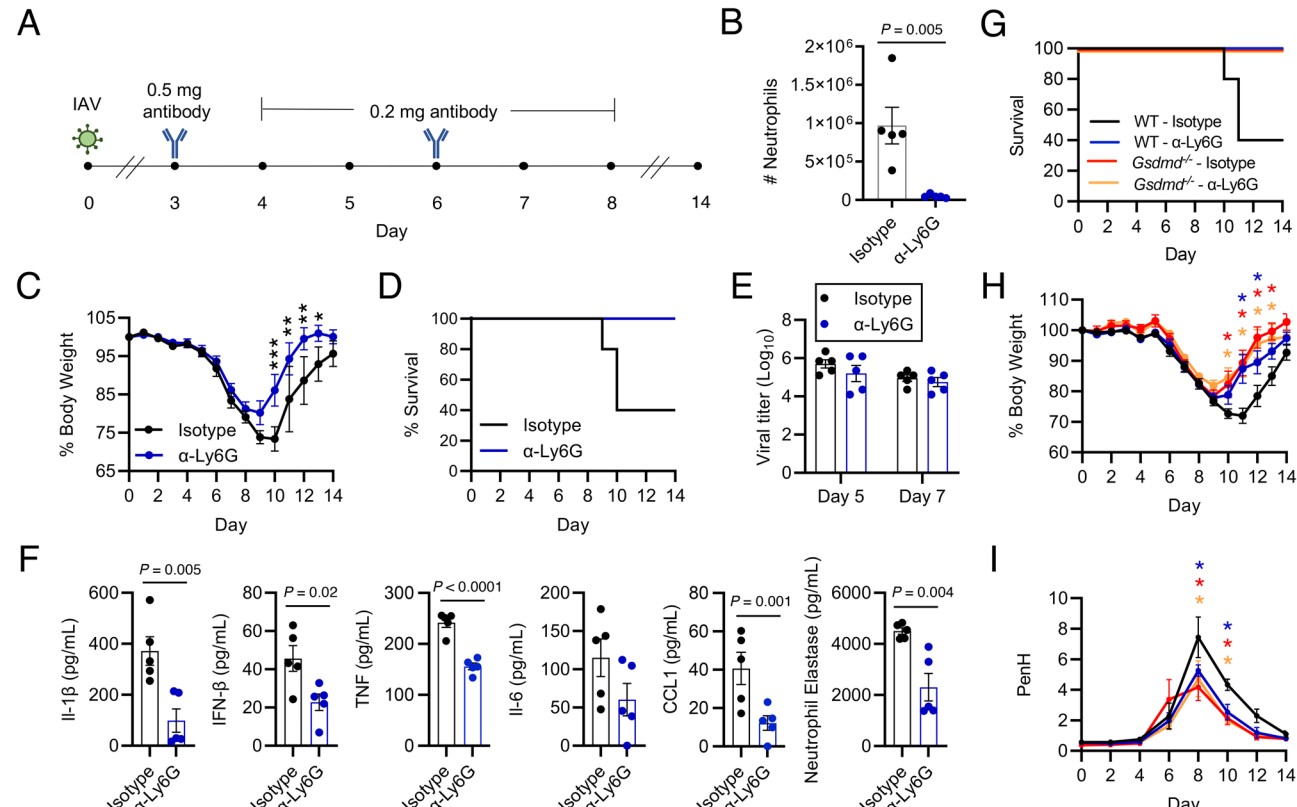

**Fig. 6 | Neutrophils increase influenza severity. A** Schematic of IAV infection (50 TCID50) and antibody injections for (**B–I**). **B** Neutrophil number in the lungs of $n = 5$ WT and $Gsdmd^{-/-}$ mice on day 5 by flow cytometry (each dot represents an individual mouse, error bars indicate SEM, p value by two-tailed unpaired t-test). **C** Weight loss of $n = 7$ mice per group (each dot is an average of individual mouse weights normalized to 100% relative to day 0, error bars indicate SEM, *$p < 0.05$, *$p < 0.01$, ***$p < 0.0001$ by two-way ANOVA followed by Bonferroni multiple comparisons test). **D** Survival analysis. **E** Viral titers from lung homogenates with $n = 5$ per group (Each dot represents an individual mouse, error bars indicate SEM, no statistically significant differences were detected comparing WT and $Gsdmd^{-/-}$ samples by two-way ANOVA followed by Tukey's multiple comparisons test). **F** ELISA measurements on lung homogenates from day 7 post infection of n = 5 per group (each dot represents an individual mouse, $p$ values determined by two-tailed

unpaired t-test). **G** Survival analysis of infected WT and $Gsdmd^{-/-}$ mice treated as in (**A**) with $n = 6$ per group. **H** Weight loss of $n = 6$ mice (each dot is an average of individual mouse weights normalized to 100% relative to day 0, error bars indicate SEM, *$p < 0.05$ by two-way ANOVA followed by Bonferroni multiple comparisons test, asterisks indicate respective color-coded comparisons to WT−Isotype). Colored lines correspond to groups as in (**G**). **I** PenH measurements by whole body plethysmography (each dot is an average of $n = 6$ mice, error bars indicate SEM, *$p < 0.05$ by two-way ANOVA followed by Bonferroni multiple comparisons test, asterisks indicate respective color-coded comparisons to WT−Isotype). Colored lines correspond to groups as in (**G**). Data in (**B–F**) represent one experiment and data in (**G–I**) represent one additional independent experiment. Source data are provided in the Source Data file.

in $Gsdmd^{-/-}$ lungs, 2) decreased gene expression associated with activated neutrophils in $Gsdmd^{-/-}$ lungs (e.g., KEGG NETosis, REACTOME degranulation), 3) decreased functionality of purified $Gsdmd^{-/-}$ neutrophils in vitro (NETosis and degranulation), and 4) demonstration that depletion of neutrophils in $Gsdmd^{-/-}$ mice did not affect IAV-induced weight loss or lung function, indicating that the neutrophils in these KO mice are non-functional. Of the immune cell types that control influenza virus infection outcomes, neutrophils are among the first to infiltrate sites of infection[40]. Beneficial functions of neutrophils include phagocytic activity, cytokine and chemokine release, and formation of neutrophil extracellular traps, which are traditionally associated with antibacterial activities but also promote an antiviral state[41–45]. Conversely, other studies have highlighted the potentially detrimental effect of unchecked neutrophil recruitment characterized by hyperinflammation and lung damage[4,46–50]. Our data add to a growing number of studies implicating excessive neutrophil activities as a driver of poor disease outcomes[46–48,50].

Although GSDMD was not well studied previously in terms of its effect on IAV replication or infection outcomes, it has been well characterized to play important roles in both macrophages and neutrophils in other contexts[51]. Indeed, pyroptotic cell death and release of IL-1 and IL-18 by macrophages are widely dependent on GSDMD in

several bacterial infections[51]. Our in vitro results with THP-1 macrophages indicated that GSDMD depletion resulted in less secretion of pro-inflammatory cytokines following infection with the IAV PR8 strain or a seasonal H3N2 strain. Neutrophils also rely on GSDMD for antibacterial responses, though GSDMD pores were reported to form primarily on granule and autophagic membranes rather than the plasma membrane in neutrophils[52]. This allowed escape of neutrophil elastase into the cytoplasm, which catalyzed noncanonical cleavage of GSDMD and further facilitated the release of IL-1β through an autophagy-dependent mechanism[52]. This unique feed-forward function of GSDMD in neutrophils may also be involved in its reported critical role in NETosis during bacterial challenges[33,53]. Our experiments also indicated that GSDMD promotes neutrophil functionality during IAV infection. While GSDMD ablation likely dampens IAV-induced inflammatory processes in both macrophages and neutrophils, our neutrophil depletion experiments implicate neutrophil activities as particularly important in mediating excessive inflammation and tissue damage in IAV infection.

IAV infection is among the most common causes of acute respiratory distress syndrome (ARDS) in adults[54]. Development of ARDS is marked by bilateral edema and worsened oxygen delivery[55], and a major contributor to these criteria is the unchecked

inflammation caused by cellular immune responses activated to control replicating virus. Specifically, neutrophil infiltration and abundance is correlated with the severity of ARDS[56], and neutrophil secretion of tissue damaging enzymes, such as neutrophil elastase and myeloperoxidase, and inflammatory extracellular traps can exacerbate progression of ARDS[54]. Likewise, NETosis promoted by GSDMD was recently identified as a driver of disease progression in an LPS-induced model of ARDS[57]. Indeed, full body *Gsdmd*−/− mice show enhanced survival in a bacterial sepsis model, and this phenotype was associated with decreased inflammation and NET formation[58]. However, in a follow-up study, neutrophil-specific deletion of GSDMD surprisingly resulted in enhanced lethality[59]. Neutrophils are often essential for clearance of bacterial infections while our data show that neutrophil depletion did not affect lung titers of IAV. Thus, while neutrophils amplify inflammation in both types of infections, the functions of neutrophils are likely not fully analogous in bacterial versus viral disease. Future research using neutrophil-specific GSDMD deletion may provide further information on the role of this protein in neutrophil responses during IAV infection.

While GSDMD is an effector protein in the pyroptosis pathway initiated by inflammasome activation, other intermediate proteins are likely contributing to the outcome of IAV infection. Indeed, Caspase-1/11 KO mice experience exacerbated acute illness and reduced adaptive immunity to influenza virus[60,61]. The requirement for specific caspases in activation of lung GSDMD during IAV infection remains to be investigated. Interestingly, neutrophil elastase has also been demonstrated to be capable of cleaving and activating GSDMD[33,62], which may explain the differing outcomes of Caspase-1/11-deficient versus *Gsdmd*−/− in IAV infections.

IL-1β and IL-18 levels were reduced in the lungs of *Gsdmd*−/− mice after infection but levels of these cytokines were not entirely dependent on GSDMD, despite the canonical role of GSDMD in IL-1β and IL-18 release[17–19]. These results suggest that there are partially redundant mechanisms for release of these cytokines during IAV infection. Indeed, these results are in accord with our published findings that IL-1β levels in the lung at day 2 post infection with SARS-CoV-2 are dependent on GSDMD, but that IL-1β release occurs in a GSDMD-independent manner at later timepoints post infection[9]. Interestingly, we found that loss of GSDMD had no significant impact on SARS-CoV-2 disease severity[9], highlighting intriguing differences in the pathological mechanisms activated by SARS-CoV-2 versus IAV that warrant further study. In addition, we discovered here that GSDMD promotes IFNβ expression and production during IAV infection, further suggesting that this effector protein potentiates pro-inflammatory responses beyond the release of canonical inflammasome-associated molecules. Attenuation of several pro-inflammatory cytokine pathways, including IFNβ, in *Gsdmd*−/− mice had no appreciable effect on virus replication. However, it is well established that the pathological, tissue-damaging effects of IFNs continue to accumulate beyond the point at which their antiviral effects peak[63–65]. Our data indicate that loss of GSDMD during IAV infection balances the inflammatory response toward a less tissue-damaging phenotype while maintaining a response sufficient to combat virus replication. Indeed, during revision of our manuscript, a report was published in which an independently generated *Gsdmd*−/− mouse line challenged with a high dose of H3N2 IAV was protected from virus-induced lethality with minimal effects on virus load compared to WT animals[26]. Minor effects of GSDMD on neutrophil recruitment to the lungs were reported[26]. Future studies will likely uncover further nuanced roles for GSDMD depending on individual virus strain pathogenicity and dosing.

Overall, we found that GSDMD promotes inflammatory gene programs, tissue-damaging neutrophil activities, and lung pathology during IAV infection. Importantly, while GSDMD exacerbates severe illness, it is dispensable for recovery from infection, including lung function, thus identifying it as a promising candidate for host-directed therapeutic targeting during IAV infection.

## Methods

Our research complies with all relevant ethical regulations and was approved by the Institutional Biosafety Committee (IBC) and Institutional Animal Care and Use Committee (IACUC) of The Ohio State University.

### Virus stocks

Influenza virus strains A/Puerto Rico/8/34 (H1N1) (PR8, provided by Dr. Thomas Moran of the Icahn School of Medicine at Mt. Sinai) and A/Victoria/361/2011 (H3N2, provided by BEI Resources, NR-44042) were propagated in 10-day embryonated chicken eggs (Charles River Laboratories) and titered as previously described[66]. Stocks of PR8-Cre[39] grown in MDCK cells were kindly provided by Dr. Ryan Langlois (University of Minnesota). All viruses were titered on MDCK cells.

### Mouse Studies

Male and Female *Gsdmd*−/− mice on the C57BL/6 J background generated by Dr. Russell Vance (UC Berkeley)[67] were purchased from Jackson Laboratories (Strain # 032663). WT control C57BL/6 J mice were also purchased from Jackson Laboratories (Strain # 000664). Female mice possessing a loxP-flanked stop codon that prevents transcription of the red fluorescent protein tdTomato were purchased from Jackson Laboratories (B6.Cg-Gt(ROSA)26Sortm14(CAG-tdTomato)Hze/J, C57BL/6 J background, strain #007914). All mice were 8–12 weeks of age at the time of experimentation. Mice were housed in the Ohio State University's Biomedical Research Tower vivarium, which is maintained at 68–76 degrees F, with a 12:12 light dark cycle, and humidity between 30–70%. Autoclaved individually ventilated cages (Allentown) were used for housing. Mice were fed irradiated natural ingredient chow ad libitum (Evnigo Teklad Diet 7912). Reverse osmosis purified water was provided through an automated rack water system. Cages included 1⁄4 inch of corn cob bedding (Bed-o-Cobs, The Andersons) with cotton square nesting material. Mice were infected intranasally under anesthesia with isoflurane according to protocols approved by the Ohio State University Institutional Animal Care and Use Committee. Mouse infections were performed with a dose of 50 TCID50 of PR8 or 250 TCID50 for PR8-Cre. The higher dose of PR8-Cre was used due to its moderate attenuation relative to PR8. For neutrophil depletion experiments, mice were treated with 0.5 mg anti-mouse Ly6G antibody (InVivoMAb, BE0075-1) or 0.5 mg rat IgG2a isotype control, anti-trinitrophenol antibody (InVivoMAb, BE0089) via intraperitoneal injection on day 3 post infection. From days 4 through 8 post infection, mice were injected with 0.2 mg of the same antibodies. Enhanced pause (PenH) values were determined via whole-body lung plethysmography (Buxco Small Animal Whole Body Plethysmography). Mice were acclimated to the plethysmography chamber for 3 days prior to infection for 10 min/day. On days 0–14 post infection, mice were put in plethysmography chamber for a 5 min acclimation followed by a 5 min reading. PenH measurements were taken every 10 s during the 5 min recording period and the PenH values for the 10 s intervals were averaged and expressed as an absolute PenH value for that day. For determining lung titers and cytokine levels, tissues were collected and homogenized in 1 ml of PBS, flash-frozen, and stored at −80 °C prior to titering on MDCK cells or analysis via ELISA. TCID50 values were calculated using the Reed-Muench method. Greater than 30% weight loss was the primary criteria used for humane endpoint euthanasia in mouse experiments. All experiments were approved by The Ohio State University IACUC under protocol #2016A00000051-R2. This protocol adheres to the guidelines set out by the NIH adopted Guide for the Care and Use of Laboratory Animals.

## Cell Culture and Infections

All cells were cultured in a humidified incubator at 37 °C with 5% CO2. Vector control and GSDMD KD THP-1 cells were generated by lentiviral shRNA-mediated targeting and were generously provided by Dr. Amal Amer at The Ohio State University. Cells were maintained in RPMI 1640 medium (Fisher Scientific) supplemented with 10% EquaFETAL bovine serum (Atlas Biologicals). THP-1 were differentiated into macrophages by treatment with 25 nM phorbol myristate acetate (PMA) for 3 days as previously described[68] to allow for differentiation into macrophages. Infection of THP-1 cells with PR8 or H3N2 viruses was done at an MOI of 10 for 48 h prior to collection of cells for Western blotting or cellular media for measurement of cytokines and LDH release. MDCK cells (BEI Resources, NR-2628) were grown in DMEM (Fisher Scientific) supplemented with 10% EquaFETAL bovine serum.

## Ex-vivo bone marrow neutrophil studies

Neutrophils were negatively selected from WT or *Gsdmd*⁻/⁻ mouse bone marrow according to the EasySep Neutrophil Enrichment Kit recommended protocol (STEMCELL Technologies, 19762). Neutrophil enrichment was confirmed via flow cytometry. Neutrophil NETosis was quantified by seeding the enriched neutrophils in a 96 well plate at 30,000 cells/well with RPMI 1640 medium (Fisher Scientific) supplemented with 4 μM ethidium homodimer-1 (Invitrogen, L3224B). These cells were infected immediately with PR8 MOI 10 for 200 min during which fluorescence intensity (excitation/emission maxima 528/617) was measured every ten minutes using a SpectraMax i3X Multi-Mode Microplate Reader (Molecular Devices). Identical plates without ethidium homodimer-1 were infected for 6 h for measurements of neutrophil elastase and myeloperoxidase in the cell supernatants.

## Western blotting

For detection of protein expression in lung tissue, samples were lysed in a 1% SDS buffer (1% SDS, 50 mM triethanolamine pH 7.4, 150 mM NaCl) containing a cocktail of phosphatase protease inhibitors (Sigma, 4906845001) and protease inhibitors (Thermo Scientific, A32965). The lysates were centrifuged at 20,000 x g for 10 min and soluble protein supernatants were used for Western blot analysis. Equal amounts of protein (30 μg) were separated by SDS-PAGE and transferred onto membranes. Membranes were blocked with 10% non-fat milk in Phosphate-buffered saline with 0.1% Tween-20 (PBST) and probed with antibodies against GSDMD (1:1000, Abcam #ab219800; clone EPR20859) and actin (1:1000, Abcam #ab3280; clone ACTN05 (C4)) with HRP-conjugated anti-rabbit IgG secondary antibody (1:10,000, Cell Signaling Technologies (CST) #7074). For Western blotting of THP-1 cells, cell lysates were harvested with RIPA buffer (150 mM NaCl; 1% IGEPAL CA-630; 0.5% sodium deoxycholate; 0.1% SDS; 50 mM Tris-HCl, pH 8.0) supplemented with Benzonase nuclease and HALT protease and phosphatase inhibitor. Equal amounts of protein (15-30 μg) were resolved by SDS-PAGE and transferred to PVDF membranes (BioRad). Membranes were blocked with 5% milk diluted in Tris-buffered saline with 0.1% Tween 20 (TBST). Primary antibody incubations were performed overnight with antibodies diluted in 5% BSA in TBST, followed by incubation with species-specific HRP-conjugated secondary antibodies in 5% milk in TBST. The following antibodies were used: Cleaved PARP (Asp214) XP Rabbit mAb (1:1000, CST #5625 S; clone D64E10), Monoclonal Anti-Influenza NP A/CA/04/09 (H1N1) pdm (1:1000, BEI Resources #NR-19868; clone 2F4), GAPDH Rabbit mAb (1:1000, CST #2118 L; clone 14C10), Cleaved GSDMD (Asp275) Rabbit mAb (1:1000, CST #36425 S; clone E7H9G), Human GSDMD (1:1000, Abcam #ab210070; clone EPR19829), Peroxidase AffiniPure Donkey Anti-Mouse IgG (H + L) (1:10,000, Jackson ImmunoResearch #715-035-150), Peroxidase AffiniPure Donkey Anti-Rabbit IgG (H + L) (1:10,000, Jackson ImmunoResearch #711-035-152). Chemiluminescent images were acquired using a ChemiDoc Touch (BioRad).

## RNA extraction and quantification of gene expression

Mouse lung tissue was homogenized in TRIzol reagent. Following separation with 1-bromo-3-chloropropane, aqueous phase was collected, and RNA was precipitated with ethanol. Precipitated RNA was isolated and cleaned using NucleoSpin RNA extraction kit (Machery-Nagel, #740955) following the manufacturer's protocol. cDNA was generated using the iScript cDNA Synthesis Kit (BioRad, #1708890) following the manufacturer's protocol. mRNA quantification was performed using iTaq Universal SYBR Green reagents (BioRad) on a CFX-384 (BioRad) instrument using *Chmp2a* as a reference gene. Primers used in this study were acquired from Integrated DNA Technologies and are listed in Supplementary Data 3.

## ELISAs

Human IL-1β, IL-6, TNF, and IFNβ and mouse CCL1, TNF, IL-18, IFNβ, IL-1β, IL-6, neutrophil elastase, and myeloperoxidase ELISAs were performed on supernatant from lung homogenates or cell culture supernatants using the respective R&D Systems Duoset ELISA kits (catalogue # DY201, DY206, DY814-05, DY845, DY410, DY7625-05, DY8234-05, DY401, DY406, DY4517-05, and DY3667, respectively) according to manufacturer's instructions.

## Flow cytometry

Mock or infected mouse lungs were homogenized in GentleMACS C tubes (Miltenyi) containing Dnase I (Sigma Aldrich) and type IV collagenase (Worthington Biochemical Corporation). Lungs were processed into single cell suspensions using GentleMACS equipment. Single-cell homogenates were blocked in CellBlox (Invitrogen) and viability stain was performed using eFluor 780 Fixable Viability Dye (Invitrogen, 65-0865-14) according to the manufacturer's instructions. Cells were washed and stained with Invitrogen antibodies for Ly6G (1:333, clone 1A8-Ly6g, ref 46–9668), CD11b (1:333. clone M1/70, ref 64–0112), CD103 (1:333, clone 2E7, ref 63–1031), CD45R (1:333, clone RA3-6B2, ref 12–0452), MHCII (1:500, clone M5/114.15.2, ref 78–5321), CD8a (1:333, clone 53-6.7, ref 58–0081), CD4 (1:333, clone GK1.5, ref M001T02R02), CD3e (1:333, clone 17A2, ref 367-0032-82), CD69 (1:333, clone H1.2F3, ref 25-0691-82), CD19 (1:333, clone eBio1D3 (1D3), ref 67-0193-82), NK1.1 (1:200, clone PK136, ref 61-5941-82), and Ly6C (1:200, clone HK1.4, ref 48-5932-82) and BioLegend antibodies for CD45.2 (1:333, clone 104, ref 109838), CD11c (1:333, clone N418, ref 117320), and Siglec F (1:100, clone S17007L, ref 155524) for 30 min on ice. Cells were fixed with BD FACS lysis buffer for 10 min at 4 °C, resuspended in PBS, and run on a Cytek Aurora instrument. Count beads (Invitrogen) were added to each sample for quantification of the total cell number. Samples were analyzed using FlowJo Software (10.8.1).

## Histology

Lung tissue samples were fixed in 10% neutral-buffered formalin at 4 °C for 24 h and then transferred to 70% ethanol. Lungs were embedded in paraffin, sectioned, stained (H&E and anti-CD45 staining), and imaged by Histowiz (Histowiz.com, Brooklyn, NY, USA). Unbiased electronic quantification of resultant images was performed using ImageJ software and the color deconvolution method[9,29,69,70].

## RNA sequencing and Functional Gene Enrichment Analysis

RNA was extracted from lung tissue using TRIzol (Invitrogen). RNA library preparation and sequencing was performed by GENEWIZ (GENEWIZ LLC./Azenta US, Inc South Plainfield, NJ, USA) with the following methods, as provided by GENEWIZ: "The RNA samples received were quantified using Qubit 2.0 Fluorometer (ThermoFisher Scientific, Waltham, MA, USA) and RNA integrity was checked using TapeStation (Agilent Technologies, Palo Alto, CA, USA). The RNA sequencing libraries were prepared using the NEBNext Ultra II RNA Library Prep Kit for Illumina using manufacturer's instructions (New England Biolabs, Ipswich, MA, USA, #E7770S). Briefly, mRNAs were initially enriched

with Oligod(T) beads. Enriched mRNAs were fragmented for 15 min at 94 °C. First strand and second strand cDNA were subsequently synthesized. cDNA fragments were end repaired and adenylated at 3'ends, and universal adapters were ligated to cDNA fragments, followed by index addition and library enrichment by PCR with limited cycles. The sequencing libraries were validated on the Agilent TapeStation (Agilent Technologies, Palo Alto, CA, USA), and quantified by using Qubit 2.0 Fluorometer (ThermoFisher Scientific, Waltham, MA, USA) as well as by quantitative PCR (KAPA Biosystems, Wilmington, MA, USA). The sequencing libraries were multiplexed and clustered onto a flowcell. After clustering, the flowcell was loaded onto the Illumina HiSeq instrument according to manufacturer's instructions. The samples were sequenced using a 2x150bp Paired End (PE) configuration. Image analysis and base calling were conducted by the HiSeq Control Software (HCS). Raw sequence data (.bcl files) generated from Illumina HiSeq was converted into fastq files and de-multiplexed using Illumina bcl2fastq 2.17 software. One mis-match was allowed for index sequence identification." Raw fastq files were processed, aligned and quantified with a HyperScale architecture developed by ROSALIND (ROSALIND, Inc. San Diego, CA, https://rosalind.bio/). Read Distribution percentages, violin plots, identity heatmaps, and sample MDS plots were generated as part of the quality control step. Statistical analyses for differential gene expression were derived using ROSALIND. Specifically, sample counts were normalized via Relative Log Expression using DESeq2 R library[71]. DEseq2 was also used to calculate fold changes and p-values. The principal component analysis, volcano plots, and heatmaps were formatted in R using numerical values provided by ROSALIND. Gene Ontology analysis was performed using the topGO R package to determine local similarities and dependencies between GO terms in order to perform Elim pruning correction. Gene set enrichment analysis (GSEA) was performed using published methods. Gene lists were ranked by expression and put into GSEA software for analysis using the GO Biological Processes database[72,73]. Network analysis was performed using Cytoscape[74]. Additional analysis was performed using the REACTOME database[75], PanglaoDB[76], the KEGG database[77], and Ingenuity Pathway Analysis (Qiagen). Functional enrichment analysis was partially performed using Enrichr[78,79].

### Reporting summary
Further information on research design is available in the Nature Portfolio Reporting Summary linked to this article.

## Data availability
RNA sequencing raw data, as well as processed gene expression matrices (raw and normalized counts), have been deposited in the NCBI GEO database under accession code GSE230656. Raw data for Western blots and graphs within the Figures and Supplementary Figs. are provided in the Source Data file. Source data are provided with this paper.

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

## Acknowledgements

Research in the Yount laboratory is supported by NIH Grants AI130110, AI151230, HL154001, HL157215, and an American Lung Association COVID-19 and Emerging Respiratory Viruses Research Award. AZ was supported by an NSF-GRFP fellowship grant. ADK and PJD were supported by an institutional T32 postdoctoral fellowship (NIH training grant AI165391). Research in the Forero laboratory is supported by NIH grant R35 GM150806. We thank Dr. Ryan Langlois (University of Minnesota) for supplying stocks of PR8-Cre.

## Author contributions

Conceptualization: A.O.A., C.C., J.M. and J.S.Y. Data Curation: A.F. and J.S.Y. Formal analysis: S.S., M.I.M., A.Z., A.S., A.D.K., R.R., E.A.H., A.F. and J.S.Y., Investigation: S.S., M.I.M., A.Z., A.S., S.L., J.E.R., A.D.K., H.B., L.Z., P.J.D., A.C.E. and E.A.H. Visualization: S.S., M.I.M., A.Z., A.S., A.D.K., A.F. and J.S.Y. Writing – Original Draft: S.S., M.I.M., A.Z., A.D.K., J.M., A.F. and J.S.Y. Supervision: J.S.Y. Project Administration: J.S.Y. Funding Acquisition: J.S.Y. Resources: A.O.A., C.C. and J.M.

## Competing interests

The authors declare no competing interests.
