## [Peer Review File · Nature Communications]

Gasdermin D promotes influenza virus-induced mortality through neutrophil amplification of inflammationREVIEWER COMMENTS

Reviewer #1 (Remarks to the Author):

Speaks et al study shows a clear role for Gasdermin D (GSDMD) in disease severity and mortality following influenza A virus infection in mice. Using GSDMD complete knockout mice, the authors show that this deficiency provides protection from death after infection with IAV. KO mice had significantly less weight loss, 100 % survival, reduced patho-histological score and reduced inflammation – despite having similar viral titers as WT mice. Importantly, specific knockdown of neutrophils by anti Ly6G Abs in WT mice recapitulated the protective finding seen in GSDMD KO. Thus the data clearly pinpoints to a role for neutrophils, and provides some significant potential for therapeutic intervention. However, I have a few points for consideration, in particular towards the lack of a causal relationship between GSDMD and neutrophil involvement.

1) Transcriptional analysis was performed on 3 samples from each group – PCA reveals one sample from KO group to be an outlier. Has further validation of key genes been performed using PCR in larger sample sizes?

2) For additional context the volcano plot in figure 3 could be annotated to highlight some of the important genes (i.e. IL-1b, TNF, CCL-1 etc) and genes that were most significantly and differentially expressed.

3) Karmaker et al (DOI: 10.1038/s41467-020-16043-9) have shown that GSDMD has different roles in macrophages and neutrophils, for clarification of the conclusions, it would be good to include a discussion of these differences or provide more rationale for investigating the impact on macrophages in vitro in humans, and how those findings might relate to the in vivo IAV model with neutrophil involvement.

4) To assess the recruitment of neutrophils in KO mice, day 7 time point was chosen. No difference in neutrophil number or freq was observed between the different mouse strains, is it possible that the kinetics of the response could be altered in the KO mice? I think it

would be informative to determine whether there are differences earlier in the response, or to confirm the findings measuring multiple timepoints.

5) Interestingly, the CD45+ cell counts from the histology revealed a modest decrease in lung infiltrate in KO mice, was a similar finding observed by flow cytometry? It was surprising to see no differences in cell populations in Sup fig 4.

6) Silva et al (<https://doi.org/10.1186/s13054-022-04062-5>) show evidence for the role of GSDMD in triggering NETosis, have the authors explored NET formation (or lack thereof) in the KO mice following influenza, or does the transcriptional pathway analysis reveal any differences in netosis or pyroptosis pathways?

7) Given the detrimental role for neutrophils in influenza infection, and the marked improvement in survival in the absence of GSDMD, can the authors provide insight into whether this finding is directly related to the inflammasome response in neutrophils, or whether it is more in-direct? Liu et al (<https://doi.org/10.1182/blood.2022016931>) recently published an article in Blood examining the role for GSDMD specifically in neutrophils during sepsis. Intriguingly, they found disease was exacerbated. This would be worth discussing.

Reviewer #2 (Remarks to the Author):

The manuscript aimed to examine the role of Gsdmd during IAV infection. While the study is interesting, the conclusions are not well supported by the data and the mechanisms involved are poorly defined. Specifically, as outlined in the points outlined below, data throughout is contradictory, and the claim neutrophil responses are diminished in the absence of Gsdmd is very weak, and not well supported by the data. Most significantly, a considerable concern is the genetic background mismatch between wildtype and Gsdmd knockout mice used in the study.

Line 363 – The GSDMD C57BL/6N knockout mice are not adequately detailed or referenced. The mice were originally generated by Feng Shao and the study should be referenced. The

genetic background of the mice used also needs to be included in the methods section.

As outlined in the submitted Reporting Summary document, C57BL/6J mice were used in the study as wildtype controls rather than appropriate C57BL/6N controls (refer to Jackson Lab website listing C57BL/6NJ mice as appropriate controls for the Gsdmd ko mice (also known as C57BL/6N) controls). C57BL/6NJ and C57BL/6J mice display differences in susceptibility to IAV (PMID: 30713529), as well as altered pulmonary inflammatory responses (PMID: 36810092). This raises considerable questions regarding the validity of the results and the entire study. It can't be ruled out that the differences seen between Wildtype and Gsdmd ko mice are not due to mismatching of genetic background.

Gsdmd mediates cell death, yet the study lacks any analysis of cell death other than Gsdmd cleavage in whole lung tissues. More in-depth analysis of cell death in different cell types is warranted.

Fig 1 – it's not clear what the euthanasia criteria was for the mice. Day 7 analysis is a late timepoint to examine innate immune responses. Can the authors provide more detailed analysis?

Fig 2 Histology – Can the authors justify why the location/area of the lung shown for the representative images are not consistent between the two genotypes?

Fig 2C-D - Gsdmd knockout mice showed reduced CD45+ cells by IHC. This data contradicts the data in Supp Fig 4, which suggests immune cell recruitment is not altered in the knockout mice. An earlier timepoint should also be included in Supp Fig 4. The data also contradicts the data in Fig 2A. The authors should also perform IHC for Ly6G in lung tissues to match Fig 2C-D. Can the authors explain why the data is contradictory?

The authors state their data 'implicated a diminished neutrophil response'. This conclusion is not well supported by the data presented. Whole lung tissues were examined by RNAseq (Fig 3). As shown in the previous figure for CD45+ staining (Fig 2D), the cellular makeup of the lung is different. Day 7 again is also a very late timepoint to be examining innate

immune responses. The RNAseq data in Fig 3 for the Ko mice is also very variable with low n numbers (n=3). Uninfected controls were missing from the experiment and would allow basal differences to be addressed. Finally, the genes in Fig 4A are not neutrophil specific and drawing conclusions about neutrophil function from whole lung tissue data is not well justified.

Fig 4D. The n numbers are low and the data is variable. Data was analysed at day 7. Analysis of an cytokines at an additional earlier timepoint would be appropriate. This data also does not support a change in neutrophil responses.

Fig 4A-F Uninfected controls are missing.

The neutrophil depletion data in Fig 5 is largely already published in PMID: 23827683 and does not inform a role for GSDMD but rather depletion of neutrophils themselves. This data should be removed. Additionally, the data does not recapitulate the phenotype in Gsdmd ko mice, as the authors state neutrophil numbers are not altered (Fig. 4F).

Supp Fig 2 – uninfected controls are missing from all panels. The PR8 strain used for the studies in human THP-1 is mouse adapted (passaged through mouse lung over 100 times) and therefore has questionable biological relevance to humans. Data showing cleaved GSDMD is missing from Supp Fig2A.

Supp Fig 3 – Whole lung tissues were processed for flow cytometry. It would be helpful to include %s on the graphs. CD45- epithelial, endothelial and fibroblast cells make up a large proportion of the lung. The authors should justify why these are missing from the 3rd graph SSC vs CD45.2. Additionally, the Ly6G vs CD19 graph has a Ly6G high population, yet this population should have been removed by the previous gating. CD3 should be CD3e. SiglecF is spelt SiglicF. Eosinophils express SiglecF yet are gated as SiglecF negative.

Lines 80-84 - reference to MG53 has little context for the reader to follow.

Lines 88-90 “GSDMD pores mediate release of specific pro-inflammatory cytokines, which

can promote leukocyte recruitment and viral clearance". Greater detail is required as to which pro-inflammatory cytokines GSDMD mediates the release of. The role of GSDMD in viral clearance is not well supported by the referenced reviews on IL-1b and IL-18.

Lines 91-93 "In vitro, GSDMD was shown to be nonessential for macrophage death induced by influenza virus infection, though effects on cytokine and chemokine responses were not examined" It needs to be noted the referenced study utilised the PR8 IAV strain, which unlike human seasonal IAV, is largely resistant to IAV infection. This detail also needs to be included in the discussion given the use of the PR8 strain in the study.

Methods - PR8 infectious dose should be also listed as PFU to allow comparison to other studies.

Can the authors justify why only female mice were used in the study? The claim that female mice are more susceptible to IAV infection (outlined in Author Summary document) is not supported by the literature. Male mice should also be included in the study.

Fig .1D - Y axis should include TCID50 not its clear the data is not PFU/lung.

Reviewer #3 (Remarks to the Author):

In the submitted manuscript, Speaks et al. demonstrate a novel role for gasdermin D in promoting pathogenesis and mortality during IAV infection in mice. They report that full body GSDMD KO mice experience attenuated weight loss, lung dysfunction/histopathology, and mortality compared to WT controls. Transcriptomics analysis of IAV-infected lungs revealed a signature consistent with loss of neutrophil function in GSDMD KO animals and neutrophil depletion improved infection outcomes in WT mice without impacting GSDMD KO animals. These results argue that neutrophils are the major drivers of GSDMD-mediated immunopathology in IAV-infected WT mice. The manuscript contributes to growing literature linking GSDMD and neutrophils with harmful inflammation and pathology in the lung. Given the high level of current interest in gasdermin biology, these findings are very timely. The data in the manuscript are clearly presented and convincing. A handful of

additional experiments to begin to illuminate the mechanisms through which GSDMD influences lung inflammation and neutrophil function in response to IAV would markedly broaden the applicability and impact of the study. Suggestions on how the authors might get there are detailed below.

Major points:

1. While the authors do a good job of implicating GSDMD and neutrophils in influenza pathogenesis, the connection between GSDMD and neutrophil function is left unresolved. Do the authors propose that loss of GSDMD has an intrinsic effect on neutrophil function or do they think the immune milieu of the lung is altered such that neutrophil activation is impacted? Additional data that can help point towards one model vs. the other would aid in the interpretation of their findings. I concede that working out the whole mechanism in vivo is beyond the scope of this study, but experiments looking at how WT v. GSDMD KO neutrophils respond to IAV in vitro should be doable. Measurements of cell death, gene expression (as in Fig. 4 but in vitro), NE and MMP expression (as in Fig. 4 but in vitro) would go a long way in helping define whether loss of GSDMD impacts neutrophils in a cell-intrinsic capacity. Another experiment could ask how WT neutrophils respond to supernatants from IAV-infected WT vs. GSDMD KO macrophages (to test the cell extrinsic/immune milieu model). These types of reductionist experiments, coupled with additional text elaborating on the multiple ways loss of GSDMD could impact neutrophil function, would significantly boost the impact of the manuscript.

2. There is seemingly a discrepancy between IHC data presented in Fig. 2 (from which the authors conclude that CD45+ cells are lower in the lungs of GSDMD KO mice in response to IAV) and Fig. S4 (from which the authors conclude that GSDMD does not affect immune cell recruitment to the lung). It is possible/likely that authors are losing CD45+ cells in flow experiments by virtue of relying on a viability dye. Measuring total cell numbers via flow instead of live cells might help address this inconsistency. Additional analysis of lung sections to enumerate and categorize neutrophils (degenerate, immature etc.), to shed light on whether certain types of neutrophils are being lost in flow cytometry experiments, would also help reconcile data in Figures 2, 4, and S4.

3. Since the infections were presumably done in parallel (as evidenced by data in Fig. G-I), were cytokines measured in the GSDMD KO–isotype and GSDMD KO–anti-Ly6G mice, as in Fig. 5F? It would be interesting to see how neutrophil depletion of the GSDMD KO mice impacts cytokine expression, even if survival and body weight are unaffected. This might even help address some of the points raised above related to the cell-intrinsic contribution of GSDMD to neutrophil function.

4. The authors reference a paper to assert that cell death during IAV infection is GSDMD-independent but it's not clear whether their THP-1 experiments follow identical parameters—that is to say, the authors should show data to rule out a role for cell death in altering cytokine expression/release in THP-1 IAV infections in S2.

Minor points:

1. In Fig. 1A, it is not clear what each lane represents. A different mouse?

2. Authors should explain why they chose Day 7 for their major readouts.

3. In line 91, elaborate what is meant by “inflammatory cytosolic components”

4. Caught a couple of typos: In line 160, it should be corresponded or correspond? In line 254, positive is spelled wrong. In line 673, cytometry is spelled wrong

REVIEWER RESPONSE

Detailed responses in blue text are below with highlights listed here:

- 1) We have added new experiments with purified neutrophils and influenza virus infection indicating that NETosis is induced upon encounter with virus and that this is facilitated by GSDMD. Moreover, we have also added new evidence of direct neutrophil infection *in vivo*. We have also added additional bioinformatic analysis of neutrophil-associated pathways within our WT and *Gsdmd*^{-/-} lung RNA sequencing results.
- 2) The issue regarding mouse background raised by Reviewer 2 stems from an error on our Reporting Summary due to copying the wrong product number from the Jackson Laboratory website where two distinct *Gsdmd*^{-/-} lines are available. We have corrected this on the Reporting Summary and provide additional information in our Methods section. Importantly, the correct WT controls were undoubtedly used in our experiments as demonstrated by animal transfer records included here.
- 3) We clarify that we chose day 7 post infection because this is within the peak of inflammation and virus replication in our model system. We have added cytokine measurements and detection of immune cells at earlier timepoints to support our choice of measurements at day 7 as the timepoint at which we see significant differences between WT and *Gsdmd*^{-/-} mice that correlate with the protection from mortality seen in the KO mice.
- 4) As requested by the journal and reviewers, we have added new data on male mice. We have now observed a protective benefit in terms of weight loss and lung function during infection of male *Gsdmd*^{-/-} mice consistent with the beneficial effect we previously saw in females.

Reviewer #1 (Remarks to the Author):

Speaks et al study shows a clear role for Gasdermin D (GSDMD) in disease severity and mortality following influenza A virus infection in mice. Using GSDMD complete knockout mice, the authors show that this deficiency provides protection from death after infection with IAV. KO mice had significantly less weight loss, 100 % survival, reduced patho-histological score and reduced inflammation – despite having similar viral titers as WT mice. Importantly, specific knockdown of neutrophils by anti ly6G Abs in WT mice recapitulated the protective finding seen in GSDMD KO. Thus the data clearly pinpoints to a role for neutrophils, and provides some significant potential for therapeutic intervention. However, I have a few points for consideration, in particular towards the lack of a causal relationship between GSDMD and neutrophil involvement.

- 1) Transcriptional analysis was performed on 3 samples from each group – PCA reveals one sample from KO group to be an outlier. Has further validation of key genes been performed using PCR in larger sample sizes?

We routinely conduct outlier detection prior to statistical analysis of gene expression across samples. Briefly, scaled expression values were imported into R for correlation-based outlier detection. Samples with Z-score >2 are considered outliers. Although one of our WT samples was found to be modestly variable, no samples met exclusion criteria for downstream analysis (see data below). This allowed us to identify 1,259 differentially expressed genes comparing infected WT and KO lungs while applying higher statistical rigor than what is commonly

acceptable for RNAseq analysis (adjusted $p < 0.01$ and fold change > 3). Nonetheless, as suggested, we performed a new *in vivo* experiment with 5 mice in each infected group and validated several key genes via qRT-PCR. These data show broadly decreased inflammatory responses following infection in *Gsdmd*^{-/-} lungs, thus confirming conclusions from our RNAseq dataset. Furthermore, we added new comparisons with mock-infected control lungs, which show that there is not a baseline difference in inflammatory states in WT versus KO animals, agreeing with cytokine measurements and immune cell flow cytometry data shown later in the manuscript.

2) For additional context the volcano plot in figure 3 could be annotated to highlight some of the important genes (i.e. IL-1b, TNF, CCL-1 etc) and genes that were most significantly and differentially expressed.

We have added annotations for several genes of interest involved in innate immunity/inflammation as requested.

3) Karmaker et al (DOI: 10.1038/s41467-020-16043-9) have shown that GSDMD has different roles in macrophages and neutrophils, for clarification of the conclusions, it would be good to include a discussion of these differences or provide more rationale for investigating the impact on macrophages *in vitro* in humans, and how those findings might relate to the *in vivo* IAV model with neutrophil involvement.

- 1) We have added a discussion of the Karmaker manuscript in which it is shown that neutrophil secretion of IL-1b requires both GSDMD cleavage and the autophagy pathway. Furthermore, we discuss their results showing that GSDMD allows cytosolic escape of neutrophil elastase that further cleaves GSDMD, making this additional feed-forward function of GSDMD unique from its known role in forming plasma membrane pores in other cell types.
- 2) We have also added more discussion regarding the published role of GSDMD in promoting NETosis (Sollberger, *Science Immunology*, 2018; Chen, *Science Immunology*, 2018).
- 3) We have also added new *in vitro* experiments with murine neutrophils showing that treatment with influenza virus induced WT neutrophil NETosis along with release of neutrophil granule components. Both responses were significantly decreased in GSDMD KO neutrophils. These results are in accord with previous studies showing direct

infection of neutrophils by influenza virus *in vitro* (Chan, *Respiratory Research*, 2020; Ivan, *Genomics*, 2013).

- 4) We have added new data confirming that neutrophils are infected by influenza virus *in vivo*.
- 5) Lastly, we have repeated and expanded experiments with WT and *GSDMD* knockdown THP1 macrophages, further confirming a role for *GSDMD* in affecting the inflammatory response to mouse-adapted and human-derived influenza viruses in macrophages. Taken together with our *in vivo* data, we propose a model in which *GSDMD* drives both neutrophil and macrophage functionality with neutrophil functions critically contributing to lung damage and amplification of inflammation.

4) To assess the recruitment of neutrophils in KO mice, day 7 time point was chosen. No difference in neutrophil number or freq was observed between the different mouse strains, is it possible that the kinetics of the response could be altered in the KO mice? I think it would be informative to determine whether there are differences earlier in the response, or to confirm the findings measuring multiple timepoints.

Day 7 is the peak of cytokine induction in our model of infection and this timepoint has been valuable for evaluating genetic differences and therapeutic options in our past studies (Kenney, *PNAS*, 2019; Kenney, *J Resp Crit Care Med*, 2021; Kenney, *Science Advances*, 2022). It is important to note that we utilize a relatively low dose of a pathogenic lineage of the H1N1 PR8 strain (Mt. Sinai New York lineage), which does not induce the immediate inflammation seen in models that use high dosing. Nevertheless, we have now added measurements of lung cytokines at day 3 and day 5 post infection, which show that cytokine induction is lower at these timepoints than at day 7 (**new Fig 4D**). Further, we note that the cytokines induced at these timepoints are not dependent on *GSDMD*. We also added measurements of immune cell populations at day 3 post infection and saw that neutrophil recruitment was minimal at this early timepoint. These results are consistent with past reports in which neutrophil recruitment in the lungs of mice peaked at day 7 following infection with several distinct influenza virus isolates (Perrone, *PLOS Pathogens*, 2008). Overall, day 7 is the timepoint at which inflammation peaks in our model and is also the timepoint at which we see measurable differences in inflammation and neutrophil products in WT versus *Gsdmd*^{-/-} mice that correlate with the increased survival of the KO animals.

5) Interestingly, the CD45+ cell counts from the histology revealed a modest decrease in lung infiltrate in KO mice, was a similar finding observed by flow cytometry? It was surprising to see no differences in cell populations in Sup fig 4.

Both flow cytometry and histology showed robust recruitment of CD45+ cells to the lungs following infection. We note that quantifications of CD45+ cells in histological analysis were marginally decreased in *Gsdmd*^{-/-} mice in only a fraction of the animals, which brought the overall average below that of WT samples. We, however, did not see a decrease in CD45+ cells when analyzing by flow cytometry whether gating on all events (as suggested by Reviewer 3) or only on viable cells. Flow cytometry provides a comparatively more reliable analysis since it is based on cells from the entire lung rather than on a single lung section. We have thus chosen to remove the CD45+ cell histology from the main text of the manuscript and now include representative images as **new Supplementary Fig 6** to accompany flow cytometry data with the general conclusion that CD45+ cells can be readily seen in both assays in the lungs of WT and *GSDMD* KO animals after infection.

We were also surprised by the result that most immune cell populations were similarly represented in WT versus *Gsdmd*^{-/-} mice. We thus repeated this experiment several times to be certain of this finding (note the high number of individual mice represented in Fig 5C day 7 timepoint). As discussed in point #4 above, day 7 is within the peak of lung inflammation in our model based on our past studies and this was confirmed by our newly added cytokine measurements on days 3 and 5 post infection. Cytokines were largely undetectable at 3 dpi and were moderately increased at 5 dpi. At these timepoints, no differences between WT and *Gsdmd*^{-/-} mice were observed, while differences were observed at day 7 (**new Fig 4D**). Further, we added analysis of immune cell infiltration at an earlier timepoint (day 3) as requested by reviewers. Minimal immune cell recruitment to the lung was evident at this timepoint and no differences between WT and *Gsdmd*^{-/-} mice were observed (**new Supplementary Fig 5**). Thus, our added timepoints support our choice of day 7 post infection for analysis of differences that correlate with the protection of GSDMD from lung damage and death during influenza virus infection. These differences support the notion that GSDMD-dependent neutrophil functions, not differences in neutrophil numbers, contribute to amplification of inflammation in the lung during influenza virus infection.

6) Silva et al (<https://doi.org/10.1186/s13054-022-04062-5>) show evidence for the role of GSDMD in triggering NETosis, have the authors explored NET formation (or lack thereof) in the KO mice following influenza, or does the transcriptional pathway analysis reveal any differences in netosis or pyroptosis pathways?

We thank the reviewer for this question. Indeed, the KEGG Pathway Database includes a gene set associated with “Neutrophil extracellular trap formation.” We found that genes in this pathway were significantly enriched in our dataset (adjusted p value = 0.0039) suggesting a decrease in NET formation in the lungs of the *Gsdmd*^{-/-} mice. A heatmap of this NET formation KEGG pathway has been added in **new Figure 5A**. This result complements our measurements of neutrophil elastase and myeloperoxidase that were also decreased in *Gsdmd*^{-/-} mice *in vivo*. Furthermore, we now show *in vitro* that WT neutrophils treated with influenza virus release DNA, neutrophil elastase, and myeloperoxidase and that these processes are blunted in the absence of GSDMD (**new Fig 5E-G**). Our *in vitro* and *in vivo* results together support a model in which infection of neutrophils triggers NETosis and release of tissue-damaging molecules that amplify inflammation and increase mortality.

7) Given the detrimental role for neutrophils in influenza infection, and the marked improvement in survival in the absence of GSDMD, can the authors provide insight into whether this finding is directly related to the inflammasome response in neutrophils, or whether it is more in-direct? Liu et al (<https://doi.org/10.1182/blood.2022016931>) recently published an article in Blood examining the role for GSDMD specifically in neutrophils during sepsis. Intriguingly, they found disease was exacerbated. This would be worth discussing.

We have added discussion of this important paper showing that neutrophil-specific deletion of GSDMD resulted in exacerbated disease in a bacterial sepsis model while full body KO of GSDMD was protective. We note that neutrophils are generally essential for clearance of bacterial infection while our data show that neutrophil depletion does not affect lung titers of influenza virus. Thus, while neutrophils amplify inflammation in bacterial and viral infections, other functions of neutrophils may not be fully analogous in bacterial versus viral disease. We do however mention in our discussion that experiments with neutrophil-specific GSDMD KO in

influenza virus infection would be a valuable future endeavor.

Reviewer #2 (Remarks to the Author):

The manuscript aimed to examine the role of Gsdmd during IAV infection. While the study is interesting, the conclusions are not well supported by the data and the mechanisms involved are poorly defined. Specifically, as outlined in the points outlined below, data throughout is contradictory, and the claim neutrophil responses are diminished in the absence of Gsdmd is very weak, and not well supported by the data. Most significantly, a considerable concern is the genetic background mismatch between wildtype and Gsdmd knockout mice used in the study.

Line 363 – The GSDMD C57BL/6N knockout mice are not adequately detailed or referenced. The mice were originally generated by Feng Shao and the study should be referenced. The genetic background of the mice used also needs to be included in the methods section.

This issue has been resolved in the text and Reporting Summary. Please see the full response to the next point below.

As outlined in the submitted Reporting Summary document, C57BL/6J mice were used in the study as wildtype controls rather than appropriate C57BL/6N controls (refer to Jackson Lab website listing C57BL/6NJ mice as appropriate controls for the Gsdmd ko mice (also known as C57BL/6N) controls). C57BL/6NJ and C57BL/6J mice display differences in susceptibility to IAV (PMID: 30713529), as well as altered pulmonary inflammatory responses (PMID: 36810092). This raises considerable questions regarding the validity of the results and the entire study. It can't be ruled out that the differences seen between Wildtype and Gsdmd ko mice are not due to mismatching of genetic background.

We thank the reviewer for noting this critical copy/paste error on our part. The mice used in our study were generated by Dr. Russel Vance and obtained from Jackson Labs. This is now highlighted in the methods with reference to Dr. Vance's original paper. We previously incorrectly copied the wrong product number corresponding to a second GSDMD KO available from Jackson Labs onto the Nature Reporting Summary. To be entirely clear, we did not use the Feng Shao animals in our study, have corrected this on the Reporting Summary, and we apologize for the confusion that this caused. We have unquestionably used the proper WT controls for our experiments. Below we provide a screenshot from the Ohio State University's mouse ordering system to show our earliest order for Vance *Gsdmd*^{-/-} animals and C57BL/6J controls.

Editorial Note: This figure has been redacted as indicated to remove third-party material where no permission to publish could be obtained and to remove confidential personal information.

Gsdmd mediates cell death, yet the study lacks any analysis of cell death other than Gsdmd cleavage in whole lung tissues. More in-depth analysis of cell death in different cell types is warranted.

While GSDMD is known to mediate cell death in macrophages and neutrophils, our study focuses on GSDMD's impact on outcomes of influenza virus infection – something that has not previously been investigated. Nonetheless, we now show that “NOD-like receptor signaling pathway” and “Neutrophil extracellular trap formation” gene signatures are down as expected in the lungs of infected *Gsdmd*^{-/-} mice as compared to WT mice (new Fig 5A). Moreover, we show *in vitro* that exposure of DNA to extracellular dye indicative of NETosis and cell death is decreased in GSDMD-deficient neutrophils infected with influenza virus compared to WT cells (new Fig 5E-G).

Fig 1 – it's not clear what the euthanasia criteria was for the mice. Day 7 analysis is a late timepoint to examine innate immune responses. Can the authors provide more detailed analysis?

The euthanasia criteria used in our experiments was 30% weight loss and this is now mentioned in the Results as well as the Methods sections. Regarding timing of our measurements, inflammatory responses continue throughout the course of infection as virus is replicating and can be amplified by positive feedback loops as tissue damage occurs. As discussed above, day 7 is within the peak of virus replication, lung dysfunction, and cytokine production, and is an ideal time to measure innate inflammatory immune responses in our infection model. In support of this, we have added cytokine measurements for days 3 and 5 post infection, which show much lower levels of cytokines than at day 7 (new Fig 4D). Further, an earlier timepoint showed less robust recruitment of neutrophils and other immune cells to the lung (new Fig 5C and D).

Perhaps most importantly, day 7 is the timepoint at which we observed measurable differences in inflammatory gene signatures, neutrophil activation gene signatures, inflammatory cytokines/chemokines, and secreted neutrophil products between WT and *Gsdmd*^{-/-} mice that correlated with the overall protection of KO animals.

Fig 2 Histology – Can the authors justify why the location/area of the lung shown for the representative images are not consistent between the two genotypes?

We have now modified our H&E histology figure to zoom in on similar regions of representative damage in WT and KO lungs (see Fig 2B). Lung damage results were confirmed by inclusion of new data from independent mice showing increased lung dysfunction in WT versus *Gsdmd*^{-/-} mice via whole body plethysmography (**new Fig 2A**).

Fig 2C-D - *Gsdmd* knockout mice showed reduced CD45+ cells by IHC. This data contradicts the data in Supp Fig 4, which suggests immune cell recruitment is not altered in the knockout mice. An earlier timepoint should also be included in Supp Fig 4. The data also contradicts the data in Fig 2A. The authors should also perform IHC for Ly6G in lung tissues to match Fig 2C-D. Can the authors explain why the data is contradictory?

We do not believe that these data are truly contradictory as the differences between WT and *Gsdmd*^{-/-} lungs in CD45 histology staining were minor and only in a subset of the KO animals. As reviewer 3 points out, this minor discrepancy may be due to inclusion of dead or dying cells in histological analysis while dead cells may be lost in the flow cytometry staining procedure and removed via viability gating in flow cytometry analysis. Additionally, since histological visualization of CD45+ cells in single lung sections is inherently more qualitative than flow cytometry, which examines the full lung, we have removed the CD45 histology figure from the main text and now provide images as **new Supp Fig 6** for the purpose of supporting the flow cytometry conclusion, i.e., that immune cell infiltration can be readily seen in both WT and *Gsdmd*^{-/-} lungs. Finally, we have added a day 3 post infection timepoint showing low, but similar, levels of immune cell recruitment in WT and *Gsdmd*^{-/-} lungs (**new Fig 5C**). Overall, using two methodologies (cytometry and imaging) we have seen that immune cell infiltration occurs in both WT and *Gsdmd*^{-/-} lungs.

The authors state their data ‘implicated a diminished neutrophil response’. This conclusion is not well supported by the data presented. Whole lung tissues were examined by RNAseq (Fig 3). As shown in the previous figure for CD45+ staining (Fig 2D), the cellular makeup of the lung is different. Day 7 again is also a very late timepoint to be examining innate immune responses. The RNAseq data in Fig 3 for the Ko mice is also very variable with low n numbers (n=3). Uninfected controls were missing from the experiment and would allow basal differences to be addressed. Finally, the genes in Fig 4A are not neutrophil specific and drawing conclusions about neutrophil function from whole lung tissue data is not well justified.

Inflammatory innate immune responses are ongoing throughout infection, particularly with influenza virus that blocks early innate immune responses *in vivo* during a prolonged “stealth phase” dependent on its immune antagonizing NS1 protein (Moltedo, *Journal of Immunology*, 2009). Day 7 was chosen because this is within the peak of virus replication, lung dysfunction, and inflammation in our model (Kenney, *PNAS*, 2019; Kenney, *J Resp Crit Care Med*, 2021; Kenney, *Science Advances*, 2022), and we now support this choice with new cytokine

measurements on days 3 and 5 post infection. Furthermore, others have shown in similar infection models that innate immune cell recruitment, including neutrophils, peaks at day 7 post influenza virus infection, making this timepoint relevant for our studies (Perrone, *PLOS Pathogens*, 2008). RNA-seq data, as discussed above, provided results with highly stringent statistical significance that would not be possible with highly variable samples. Furthermore, as requested by reviewer 1, we performed qRT-PCR on an independent sample set to validate RNA-seq data (**new Fig 3C**). These samples also included mock control groups that did not show a difference in baseline gene expression between WT and GSDMD KO lungs for the panel of inflammatory genes examined. Our RNA-seq analysis now includes figures with statistically significant differences in GO Neutrophil Chemotaxis, REACTOME Neutrophil Degranulation, and KEGG Neutrophil Extracellular Trap Formation gene signatures in WT versus GSDMD KO lungs post infection (Fig 4B, **new Fig 5A and B**). We additionally measured neutrophil elastase and myeloperoxidase levels in lungs to support the conclusion that there is a defect in neutrophil functionality in *Gsdmd*^{-/-} lungs (Fig 5D).

Fig 4D. The n numbers are low and the data is variable. Data was analysed at day 7. Analysis of an cytokines at an additional earlier timepoint would be appropriate. This data also does not support a change in neutrophil responses.

We have added cytokine measurements at day 3 and day 5 post infection and have increased the n number at day 7. The new data strengthen our conclusions and resulted in statistical significance for WT versus *Gsdmd*^{-/-} comparisons for additional cytokines at day 7 (**new Fig 4D**).

Fig 4A-F Uninfected controls are missing.

We now include mock controls for our cytokine/chemokine and neutrophil product measurements (**new Fig 4D and new Fig 5D and G**). These controls show that there is no difference between WT and *Gsdmd*^{-/-} mice at baseline. We have also performed qRT-PCR for select inflammatory genes on lung samples from new groups of experimental animals that included mock controls (**new Fig 4C**). Results from these experiments confirm our RNA sequencing conclusions and show no baseline differences between WT and KO samples.

The neutrophil depletion data in Fig 5 is largely already published in PMID: 23827683 and does not inform a role for GSDMD but rather depletion of neutrophils themselves. This data should be removed. Additionally, the data does not recapitulate the phenotype in *Gsdmd* ko mice, as the authors state neutrophil numbers are not altered (Fig. 4F).

We thank the reviewer for directing us to this interesting publication, which we now cite in our manuscript. Brandes, et al. use distinct neutrophil depletion strategies (timing, number, and magnitude of antibody doses) from that used in our manuscript. Nonetheless, their neutrophil depletion regimens prior to infection were beneficial to infection outcome. The results obtained in our study are thus unique but also in accord with those obtained by Brandes, et al.

As per the reviewer's comment regarding phenotype recapitulation, we have modified our text to more carefully state that our neutrophil depletion regimen specifically recapitulated the survival benefit, decreased inflammation, and lung function benefit seen in our *Gsdmd*^{-/-} mice while similarly showing no effect on viral titers. While neutrophils are not depleted in our *Gsdmd*^{-/-} mice, our results overall support a link between decreased neutrophil activity, whether via

GSDMD KO or neutrophil depletion, and decreased inflammation and death during IAV infection.

Supp Fig 2 – uninfected controls are missing from all panels. The PR8 strain used for the studies in human THP-1 is mouse adapted (passaged through mouse lung over 100 times) and therefore has questionable biological relevance to humans. Data showing cleaved GSDMD is missing from Supp Fig2A.

We have added new THP-1 cell data with mock controls and with human seasonal influenza virus (**new Supp Fig 3**). Our conclusion remains that inflammatory responses to influenza virus are diminished in the absence of GSDMD.

Supp Fig 3 – Whole lung tissues were processed for flow cytometry. It would be helpful to include %s on the graphs. CD45- epithelial, endothelial and fibroblast cells make up a large proportion of the lung. The authors should justify why these are missing from the 3rd graph SSC vs CD45.2. Additionally, the Ly6G vs CD19 graph has a Ly6G high population, yet this population should have been removed by the previous gating. CD3 should be CD3e. SiglecF is spelt SiglicF. Eosinophils express SiglecF yet are gated as SiglecF negative.

We have now included percentages on flow cytometry plots depicting our gating strategies. As for CD45- cells, our lung dissociation methods utilizing collagenase and DNase preferentially release CD45+ immune cells for analysis. We have found that other methods, such as using Dispase digestion, allows for greater numbers of CD45- cells to be obtained. Since our goal here was to examine immune cells, we believe our methodologies are appropriate and our relative lack of CD45- cells is not surprising. We have also addressed the issue regarding the Ly6G vs CD16 graph as this had been inadvertently shifted to the right in the previous version of the figure. Additionally, CD3 has been changed to CD3e and SiglecF is now spelled correctly. Lastly, we are not gating on SiglecF negative cells for eosinophils, but rather the gated cells are expressing lower levels of SiglecF than alveolar macrophages. We thank the reviewer for carefully reviewing this figure and identifying our errors.

Lines 80-84 - reference to MG53 has little context for the reader to follow.

We have added further description of MG53. Our work with MG53 during influenza virus infection implicated potential downmodulation of inflammasome functions that correlated with lung protection, and thus provides context for our group's interest in roles of GSDMD during this viral infection.

Lines 88-90 “GSDMD pores mediate release of specific pro-inflammatory cytokines, which can promote leukocyte recruitment and viral clearance”. Greater detail is required as to which pro-inflammatory cytokines GSDMD mediates the release of. The role of GSDMD in viral clearance is not well supported by the referenced reviews on IL-1b and IL-18.

We have amended the sentence to state that GSDMD pores specifically mediate release of IL-1b and IL-18. IL-1 has a reported role in promoting viral clearance for which we have added the following reference: Orzalli, et al., “An antiviral branch of the IL-1 signaling pathway restricts immune-evasive virus replication,” *Molecular Cell*, 2018.

Lines 91-93 “In vitro, GSDMD was shown to be nonessential for macrophage death induced by influenza virus infection, though effects on cytokine and chemokine responses were not examined” It needs to be noted the referenced study utilized the PR8 IAV strain, which unlike human seasonal IAV, is largely resistant to IAV infection. This detail also needs to be included in the discussion given the use of the PR8 strain in the study.

We now remark that this study utilized PR8 and that it is important to note that there may be differences in activation of cell death pathways between the PR8 strain and circulating seasonal influenza viruses.

Methods - PR8 infectious dose should be also listed as PFU to allow comparison to other studies.

TCID₅₀ titers are widely accepted by the field of virology. Multiplying the TCID₅₀ titer by 0.7 is a commonly used estimate for conversion to PFU.

Can the authors justify why only female mice were used in the study? The claim that female mice are more susceptible to IAV infection (outlined in Author Summary document) is not supported by the literature. Male mice should also be included in the study.

We have added new data with male mice. As expected, male mice showed an overall less severe weight loss than female mice and no lethality at the dose in which 60% of female mice died (**new Supplementary Fig 1**). Importantly, the absence of GSDMD in male mice has a similar protective effect during influenza virus infection in terms of average weight loss and lung dysfunction to that seen in female mice. Overall, we confirmed that our phenotype of interest is preserved in male mice. We used female mice in subsequent mechanistic studies to have a maximal dynamic range for improvement in outcome, including protection from death. We have also added a reference supporting our use of female mice from the laboratory of Dr. Sabra Klein (Johns Hopkins University), one of the world’s foremost experts on sex differences in infections, in which it was demonstrated that male mice experience less severe PR8 infections than female mice (Robinson, *PLOS Pathogens*, 2011).

Fig .1D - Y axis should include TCID₅₀ not its clear the data is not PFU/lung.

We have changed the axis for viral titers to read “Lung Viral Titer (Log₁₀ TCID₅₀/mL).”

Reviewer #3 (Remarks to the Author):

In the submitted manuscript, Speaks et al. demonstrate a novel role for gasdermin D in promoting pathogenesis and mortality during IAV infection in mice. They report that full body GSDMD KO mice experience attenuated weight loss, lung dysfunction/histopathology, and mortality compared to WT controls. Transcriptomics analysis of IAV-infected lungs revealed a signature consistent with loss of neutrophil function in GSDMD KO animals and neutrophil depletion improved infection outcomes in WT mice without impacting GSDMD KO animals.

These results argue that neutrophils are the major drivers of GSDMD-mediated immunopathology in IAV-infected WT mice. The manuscript contributes to growing literature linking GSDMD and neutrophils with harmful inflammation and pathology in the lung. Given the high level of current interest in gasdermin biology, these findings are very timely. The data in the manuscript are clearly presented and convincing. A handful of additional experiments to begin to illuminate the mechanisms through which GSDMD influences lung inflammation and neutrophil function in response to IAV would markedly broaden the applicability and impact of the study. Suggestions on how the authors might get there are detailed below.

Major points:

1. While the authors do a good job of implicating GSDMD and neutrophils in influenza pathogenesis, the connection between GSDMD and neutrophil function is left unresolved. Do the authors propose that loss of GSDMD has an intrinsic effect on neutrophil function or do they think the immune milieu of the lung is altered such that neutrophil activation is impacted? Additional data that can help point towards one model vs. the other would aid in the interpretation of their findings. I concede that working out the whole mechanism *in vivo* is beyond the scope of this study, but experiments looking at how WT v. GSDMD KO neutrophils respond to IAV *in vitro* should be doable. Measurements of cell death, gene expression (as in Fig. 4 but *in vitro*), NE and MMP expression (as in Fig. 4 but *in vitro*) would go a long way in helping define whether loss of GSDMD impacts neutrophils in a cell-intrinsic capacity. Another experiment could ask how WT neutrophils respond to supernatants from IAV-infected WT vs. GSDMD KO macrophages (to test the cell extrinsic/immune milieu model). These types of reductionist experiments, coupled with additional text elaborating on the multiple ways loss of GSDMD could impact neutrophil function, would significantly boost the impact of the manuscript.

We have added new data to address this important point:

- 1) We infected purified neutrophils with influenza virus *in vitro* and observed DNA release indicative of NETosis occurring in a GSDMD-dependent manner (**new Fig 5E and F**). We further confirmed greater WT neutrophil activation by measuring neutrophil elastase and myeloperoxidase released in the cell supernatants (**new Figure 5G**). These results indicate that neutrophils exposed to influenza virus release inflammatory products associated with NETosis and that this is facilitated by GSDMD.
- 2) We show that lung neutrophils are directly infected by influenza virus. For this, we utilized influenza virus expressing Cre recombinase in mice possessing a floxed TdTomato reporter allele. Use of this methodology confirmed previous reports of neutrophil infections *in vivo* detected with antibody staining (Hufford, *PLOS One*, 2012). We newly observed that influenza virus-infected neutrophils upregulate MHC II on their surface. In examination of MHC II on neutrophils in infected WT versus *Gsdmd*^{-/-} lungs, we observed GSDMD-independent neutrophil upregulation of MHC II during infection (**new Supplementary Figs 8 and 9**). Our data thus supports a model in which direct infection of neutrophils contributes to their activation during influenza virus infection and identifies both GSDMD-dependent and -independent effects.

3) We show additional analysis of our lung RNA sequencing data demonstrating that the KEGG pathways “Neutrophil extracellular trap formation” and “NOD-like receptor signaling pathway” are significantly altered *in vivo* comparing WT and *Gsdmd*^{-/-} mice (p values = 3.8 x 10⁻³ and 4.9 x 10⁻⁷, respectively) (new Figure 5A).

2. There is seemingly a discrepancy between IHC data presented in Fig. 2 (from which the authors conclude that CD45⁺ cells are lower in the lungs of GSDMD KO mice in response to IAV) and Fig. S4 (from which the authors conclude that GSDMD does not affect immune cell recruitment to the lung). It is possible/likely that authors are losing CD45⁺ cells in flow experiments by virtue of relying on a viability dye. Measuring total cell numbers via flow instead of live cells might help address this inconsistency. Additional analysis of lung sections to enumerate and categorize neutrophils (degenerate, immature etc.), to shed light on whether certain types of neutrophils are being lost in flow cytometry experiments, would also help reconcile data in Figures 2, 4, and S4.

Eliminating the use of the viability dye in flow cytometry analysis did not change our results, i.e., no differences in immune cell recruitment was observed comparing WT and *Gsdmd*^{-/-} samples. We include these data here (figure below) for reviewer evaluation. Also, as discussed above, we removed the CD45 IHC data from the main text and now include qualitative images as Supplementary Fig 6 with the primary conclusion from these images being that CD45⁺ cells can be visualized in both WT and *Gsdmd*^{-/-} lungs following infection.

3. Since the infections were presumably done in parallel (as evidenced by data in Fig. G-I), were cytokines measured in the GSDMD KO–isotype and GSDMD KO–anti-Ly6G mice, as in Fig. 5F? It would be interesting to see how neutrophil depletion of the GSDMD KO mice impacts cytokine expression, even if survival and body weight are unaffected. This might even help address some of the points raised above related to the cell-intrinsic contribution of GSDMD to neutrophil function.

Neutrophil depletion alone results in a profound decrease in inflammatory cytokine levels in infected lungs. We therefore did not anticipate that combining this with GSDMD KO would cause a further decrease in these levels that would be readily measurable. Thus, in these large and expensive antibody experiments, we instead utilized the mice for lung function (plethysmography) and weight loss measurements as these readouts did not require sacrificing mice and had sufficient dynamic range that would have allowed potential improvements in outcomes to be observed when combining the KO with neutrophil depletion. Overall, we did not observe any indication that there was an additive effect, thus supporting a model in which GSDMD-dependent neutrophil functions are detrimental to outcomes of influenza virus infection.

4. The authors reference a paper to assert that cell death during IAV infection is GSDMD-

independent but it's not clear whether their THP-1 experiments follow identical parameters—that is to say, the authors should show data to rule out a role for cell death in altering cytokine expression/release in THP-1 IAV infections in S2.

We have repeated THP1 experiments with PR8 infection and added new experiments with a more recent human-isolated influenza virus strain. Inflammatory cytokine secretion was reduced in *GSDMD* KD cell supernatants in both infections. We found that lactate dehydrogenase levels were decreased in the *GSDMD* KD supernatants though cleaved PARP1 could be detected in both WT and KD cells. These results together suggest that non-pyroptotic mechanisms of THP1 cell death may predominate in the absence of *GSDMD*.

Minor points:

1. In Fig. 1A, it is not clear what each lane represents. A different mouse?

Yes, each lane represents lung lysate from an individual mouse. We have now made this clear in the figure legend.

2. Authors should explain why they chose Day 7 for their major readouts.

As described above, we chose day 7 because this is within the peak of virus replication and inflammation in our mouse model. This is supported by new data that we have added to the manuscript showing lower levels of cytokines at days 3 and 5 post infection and peak lung dysfunction at day 7. Further, day 7 is the timepoint at which we were able to observe differences in inflammatory signatures and neutrophil products that correlated with protection of *Gsdmd*^{-/-} animals from death. We have added text and data to this effect justifying the choice of day 7.

3. In line 91, elaborate what is meant by “inflammatory cytosolic components”

Release of cytosolic components beyond IL-1 and IL-18 cytokines by *GSDMD* pores is less well established and we have thus deleted this statement.

4. Caught a couple of typos: In line 160, it should be corresponded or correspond? In line 254, positive is spelled wrong. In line 673, cytometry is spelled wrong

We thank the reviewer for pointing out these typos, which have now been corrected.

REVIEWER COMMENTS

Reviewer #1 (Remarks to the Author):

The authors addressed all of my comments and report a really nice study!

Reviewer #2 (Remarks to the Author):

Major

Unfortunately, this study now lacks novelty as it was recently published that GSDMD knockouts are protected from severe IAV infection (PMID: 37945599). This study needs to be cited in the introduction and discussion.

In the original version, Figure 2 showed H&E staining of lung tissues as well as IHC staining for CD45+ cells. Both of these measures showed reduced inflammation and cell infiltrates in GSDMD knockout mice. Note the CD45 analysis had a p value of <0.01 (ie 2 stars). The authors claim a p value of <0.01 is a marginal decrease. The authors have inappropriately removed the CD45 IHC quantification and changed the representative image for the GSDMD KOs to elude that there was no difference. Additionally, the flow cytometry data for cell infiltrates still contradicts the data obtained from lung tissue sections. The flow cytometry data also contradicts what is shown in Supp Fig 11 which suggests there is a difference in neutrophils in WT vs KO mice treated with isotype control antibodies. Total CD45+ counts are also missing from the flow cytometry analysis. The RNAseq data also contradicts the flow cytometry results, which suggest no difference in neutrophil infiltrates. Yet the authors state that there was an 'observed enrichment of biological pathways involved in neutrophil chemotaxis in WT versus Gsdmd^{-/-} samples as identified by GO Biological Process analysis'. This reviewer has now lost confidence in the data and the manuscripts conclusions.

Supp Fig 1 – infection of male mice. Survival data is missing and it should be shown to illustrate no male wildtype or KO mice were euthanised ie 100% survival. This data is not referenced in the results section in conjunction with the existing data in Fig 1B-C. New data

for male mice in Supp Fig 1 should all be moved to Fig 1 and the data from male and female mice pooled together. The authors conclusions that the results demonstrate 'a profound protective effect when GSDMD is absent' are not justified given that no difference was seen in male mice. As above, this reviewer has now lost confidence in the data and the manuscripts conclusions.

As previously mentioned, GSDMD plays a major role in cell death, yet cell death was not examined in vivo. The authors have failed to address this comment.

As previously mentioned, data in Fig 5 is largely already published in PMID: 23827683 and does not inform a role for GSDMD but rather depletion of neutrophils themselves. This data should be removed especially as neutrophil numbers are not altered in GSDMD ko mice. Therefore, data in Fig 5 does not support the findings in GSDMD ko mice and should be removed.

Minor:

Fig 4A. The gene names are too small to read.

Supp Fig 3A. Analysis of cleaved GSDMD is missing. Molecular weights are also missing.

Supp Figure 4. IAV infection can alter macrophage and neutrophil properties. This is evident by the change in SSC. The appropriate control to set the tdTomato gate is lacking i.e., mice infected with IAV that does not express Cre.

Reviewer #3 (Remarks to the Author):

The authors have gone above and beyond in responding to reviewer's critiques. I support its publication.

REVIEWER RESPONSE

We were pleased to see the strongly positive comments from Reviewers 1 and 3 who remarked, **“The authors addressed all of my comments and report a really nice study!”** and **“The authors have gone above and beyond in responding to reviewer’s critiques. I support its publication.”** Below we address remaining concerns from Reviewer 2.

Reviewer 2

Unfortunately, this study now lacks novelty as it was recently published that GSDMD knockouts are protected from severe IAV infection (PMID: 37945599). This study needs to be cited in the introduction and discussion.

We appreciate the *Nature Communications* policy that states “At *Nature Communications*, we commit to disregard from our editorial evaluation any competing works that are published while a submission to our journal is under review or under revision by the authors.”

We note that PMID: 37945599 published by Michelle Tate’s group was submitted and published during our review/revision period and that our manuscript was submitted to *Nature Communications* and posted as a bioRxiv preprint two months prior to the original reported submission of PMID: 37945599.

We have now cited the published manuscript in our Introduction and Discussion. We note in the Discussion that despite the differences in the kinetics of inflammation and immune cell infiltration of the Tate group’s high dose H3N2 infection compared to our low dose H1N1 infection, the overall conclusion of PMID: 37945599 is in line with our conclusion that GSDMD promotes influenza virus-induced inflammation and pathogenesis. Thus, this published work supports the broad applicability and soundness of our work. The minor distinctions in our results compared to the Tate paper also open new future avenues of research aimed at understanding nuances of GSDMD-dependent effects in infections with different viral strains and doses.

In the original version, Figure 2 showed H&E staining of lung tissues as well as IHC staining for CD45+ cells. Both of these measures showed reduced inflammation and cell infiltrates in GSDMD knockout mice. Note the CD45 analysis had a p value of <0.01 (ie 2 stars). The authors claim a p value of <0.01 is a marginal decrease. The authors have inappropriately removed the CD45 IHC quantification and changed the representative image for the GSDMD KOs to elude that there was no difference. Additionally, the flow cytometry data for cell infiltrates still contradicts the data obtained from lung tissue sections. The flow cytometry data also contradicts what is shown in Supp Fig 11 which suggests there is a difference in neutrophils in WT vs KO mice treated with isotype control antibodies. Total CD45+ counts are also missing from the flow cytometry analysis. The RNAseq data also contradicts the flow cytometry results, which suggest no difference in neutrophil infiltrates. Yet the authors state that there was an ‘observed enrichment of biological pathways involved in neutrophil chemotaxis in WT versus *Gsdmd*^{-/-} samples as identified by GO Biological Process analysis’. This reviewer has now lost confidence in the data and the manuscripts conclusions.

We visualized the presence of CD45+ immune cells via IHC in the lungs of all WT and *Gsdmd*^{-/-} animals following infection. As we noted previously, we no longer include the quantification of IHC images of CD45+ staining because examination of a single lung section is inherently less reliable than flow cytometry on cells from the entire lung. Indeed, flow cytometry showed no major differences in infiltration of individual immune cell subsets in WT versus *Gsdmd*^{-/-} lungs. **We have now included the overall CD45+ cell counts from these experiments as requested by the reviewer (new Supp Fig 6A).** These data show recruitment of CD45+ immune cells to WT and *Gsdmd*^{-/-} lungs. Thus, the qualitative IHC images showing CD45+ cells present in both WT and *Gsdmd*^{-/-} lungs are in accord with flow cytometry quantifications.

Supplemental Figure 11 is a representative “spot check” of animals to confirm neutrophil depletion in WT versus *Gsdmd*^{-/-} animals. The isotype control animals showed neutrophils present after infection in both WT and KO mice with neutrophil percentages being within the expected range of variability for our *in vivo* studies. The values do not indicate a decrease of neutrophils in KO animals. Importantly, the focus of this figure, neutrophil depletion, was highly effective in both WT and *Gsdmd*^{-/-} mice as expected.

The reviewer rightly suggests that a decrease in expression of genes associated with the GO Biological Process “Neutrophil Chemotaxis” in *Gsdmd*^{-/-} mice might be expected to correlate with a decrease in neutrophil recruitment. However, neutrophil numbers measured by flow cytometry were similar in WT versus *Gsdmd*^{-/-} lungs following infection. While this may seem contradictory at first glance, it is important to note that neutrophils are recruited to inflammatory sites via many redundant mechanisms. Further, while the genes are inferred from GO term analysis to be associated with neutrophil migration, upon deeper inspection, they also represent cytokine, chemokine, and receptor genes that are upregulated upon neutrophil activation (PMID: 32719519). Likewise, other gene sets associated with neutrophil functionality, such as KEGG NETosis and REACTOME Neutrophil Degranulation, are also decreased in KO lungs. Our data thus indicate that neutrophil functionality, not recruitment, is decreased in the absence of GSDMD. We support this interpretation with ELISA measurements of neutrophil elastase and myeloperoxidase, which are released by activated neutrophils and are indeed decreased in *Gsdmd*^{-/-} lungs. We further support our conclusions with *in vitro* experiments showing that neutrophil NETosis and enzyme release were impaired upon infection of *Gsdmd*^{-/-} neutrophils. Thus, the apparent contradiction is resolved by a deeper consideration of the data beyond the name of the GO term, particularly when examined in context with our full dataset. **We have made minor changes to the text of the Results section to guide readers, such as the addition of the statement “These results may suggest a decrease in neutrophil numbers or decreased activation status of recruited neutrophils³⁰.”**

Supp Fig 1 – infection of male mice. Survival data is missing and it should be shown to illustrate no male wildtype or KO mice were euthanised ie 100% survival. This data is not referenced in the results section in conjunction with the existing data in Fig 1B-C. New data for male mice in Supp Fig 1 should all be moved to Fig 1 and the data from male and female mice pooled together. The authors conclusions that the results demonstrate ‘a profound protective effect when GSDMD is absent’ are not justified given that no difference was seen in male mice. As above, this reviewer has now lost confidence in the data and the manuscripts conclusions.

We note in the results section that infection of male mice resulted in “no fatalities.” **We have added a survival graph (new Supp Fig 1A).**

Presenting the data for male and female mice in a disaggregated format is appropriate in this case due to the well characterized sex differences in influenza virus infection severity in which females experience significantly more severe infections (PMID: 21829352). Furthermore, *Nature* journal policies call for sex disaggregation of data when possible (PMID: 35585338).

Male *Gsdmd*^{-/-} mice showed statistically significant improvements in lung function compared to WT mice during infection. Thus, the protective phenotype afforded by loss of GSDMD was seen in both male and female mice. **We have softened our language by deleting the word “profound” in the statement mentioned by the reviewer.**

As previously mentioned, GSDMD plays a major role in cell death, yet cell death was not examined *in vivo*. The authors have failed to address this comment.

Figure 1A shows that GSDMD cleavage, indicative of pyroptosis, occurs in the mouse lung upon influenza virus infection. Genes associated with neutrophil NETosis, a terminal process linked to cell death, were decreased in *Gsdmd*^{-/-} lungs. Further, we examined neutrophil NETosis *in vitro* upon influenza virus infection and found that this was decreased in the absence of GSDMD. We also examined cell death in macrophages *in vitro* and found that GSDMD deficiency did not prevent activation of death pathways in this cell type upon infection. Overall, we have made a reasonable effort to address this comment.

As previously mentioned, data in Fig 5 is largely already published in PMID: 23827683 and does not inform a role for GSDMD but rather depletion of neutrophils themselves. This data should be removed especially as neutrophil numbers are not altered in GSDMD ko mice. Therefore, data in Fig 5 does not support the findings in GSDMD ko mice and should be removed.

The regimen for neutrophil depletion used in PMID: 23827683 differed substantially from that used in our manuscript in terms of timing, magnitude of antibody doses, and number of antibody doses. We also measured different inflammatory outcomes and lung functionality that is absent from the published work. Thus, our experiments are both distinct and complementary to those published in PMID: 23827683.

We recognize that neutrophil depletion and genetic ablation of GSDMD are not equivalent, but rather, we note that they phenocopy each other in influenza virus infection outcomes in terms of decreased weight loss, improved survival, improved lung function, and decreased lung inflammation, each without affecting viral titers. While this does not demonstrate that GSDMD acts through neutrophils, it is consistent with this model. Additional data in support of this model include: 1) decreased levels of neutrophil elastase and myeloperoxidase (tissue damaging enzymes released by neutrophil degranulation) in *Gsdmd*^{-/-} lungs, 2) decreased gene expression associated with activated neutrophils in *Gsdmd*^{-/-} lungs (NETosis, degranulation), 3) decreased functionality of purified *Gsdmd*^{-/-} neutrophils *in vitro* (NETosis and degranulation), and 4) demonstration that depletion of neutrophils in *Gsdmd*^{-/-} mice did not affect influenza virus-induced weight loss or lung function, indicating that the neutrophils in these KO mice are non-functional. **We have added new sentences in our Discussion section that synthesizes the individual pieces of evidence that, in their totality, support a model wherein GSDMD is required for neutrophil functionality that amplifies lung inflammation and damage during influenza virus infection.**

We have also made a good faith effort in considering potential neutrophil-specific GSDMD knockout strategies that could further supplement our conclusions, but have not had success in identifying feasible experiments:

1. When considering adoptive transfer experiments, we determined that such experiments were unlikely to yield interpretable results given the short-lived nature of neutrophils (6-8 h half-life) and long course of influenza virus infection (2 weeks) coupled with the need to deplete the vast numbers of endogenous neutrophils without targeting the injected cells.
2. *Gsdmd*-flox mice would provide a means to specifically examine roles of GSDMD in neutrophils. We have inquired with more than 15 groups in the US and abroad that either published with these mice or that we speculated could potentially have them. We have not received a response from anyone possessing these mice. Additionally, we note that the most published *Gsdmd*-flox mouse is on the C57BL/6N mouse background and would need to be backcrossed 10 times to the 6J background to be compared with our current study (note that Reviewer 2 previously commented in round 1 of our review that 6N mice have altered neutrophil responses compared with 6J mice). Thus, even if able to obtain these mice, the intervals required for breeding, crossing and expanding the colony is not feasible for the timely publication of the current manuscript on this topic in this highly competitive field.

Minor:

Fig 4A. The gene names are too small to read.

We have newly provided a Supplementary Table (new Supp Table 2A,B) listing the gene names and expression data used for generating these heat maps.

Supp Fig 3A. Analysis of cleaved GSDMD is missing. Molecular weights are also missing.

We have added molecular weight indicators. We have also added the requested data on cleaved GSDMD from these experiments.

Supp Figure 4. IAV infection can alter macrophage and neutrophil properties. This is evident by the change in SSC. The appropriate control to set the tdTomato gate is lacking i.e., mice infected with IAV that does not express Cre.

This suggestion would provide an interesting control for future experiments. Given that 1) the direct infection of neutrophils *in vivo* is supported by literature precedent, 2) we also see MHCII upregulation on neutrophils in WT and *Gsdmd*^{-/-} mice lacking the tdTomato allele, 3) this is not a major focus of the manuscript, we have maintained this supplementary figure.

REVIEWERS' COMMENTS

Reviewer #1 (Remarks to the Author):

The authors have provided thorough and justified responses to the reviewers comments.